# Provably sample-efficient RL with side information about latent dynamics

**Yao Liu**[*]
Amazon Web Services
yaoliuai@amazon.com

**Dipendra Misra**
Microsoft Research
dipendra.misra@microsoft.com

**Miroslav Dudík**
Microsoft Research
mdudik@microsoft.com

**Robert E. Schapire**
Microsoft Research
schapire@microsoft.com

## Abstract

We study reinforcement learning (RL) in settings where observations are high-dimensional, but where an RL agent has access to abstract knowledge about the structure of the state space, as is the case, for example, when a robot is tasked to go to a specific room in a building using observations from its camera, while having access to the floor plan. We formalize this setting as transfer reinforcement learning from an *abstract simulator*, which we assume is deterministic (such as a simple model of moving around the floor plan), but which is only required to capture the target domain's latent-state dynamics approximately up to *unknown* (bounded) perturbations (to account for environment stochasticity). Crucially, we assume no prior knowledge about the structure of observations in the target domain except that they can be used to identify the latent states (but the decoding map is unknown). Under these assumptions, we present an algorithm, called `TASID`, that learns a robust policy in the target domain, with sample complexity that is polynomial in the horizon, and *independent* of the number of states, which is not possible without access to some prior knowledge. In synthetic experiments, we verify various properties of our algorithm and compare it with several transfer RL algorithms that require access to "full simulators" (i.e., those that also simulate observations).

## 1 Introduction

When learning from scratch, reinforcement learning (RL) in the real world can be very expensive. For example, a robot learning to navigate in a building might need to explore every possible state or location, which can be painstakingly costly and time-consuming. Sometimes, however, it is not necessary to begin such a learning process from scratch. For instance, in the robot example, we might have access to a general floor map of the building. How can this kind of high-level but imprecise information be used to learn how to operate in the environment more quickly and more effectively?

In this paper, we study how to effectively leverage prior information in the form of such "abstract" descriptions of the environment. We formalize this abstract description as an "abstract simulator," which, like a map, can be used as an imperfect model. Importantly, our abstract simulators differ from more standard simulators in that they only focus on the "latent structure" of the environment dynamics, not on the observations that might be experienced by an agent in the environment.

In general, fully faithful simulators of even the latent dynamics might be hard to build, for instance, due to the difficulty of exactly modeling all probabilistic outcomes, as when the robot's actions do

---

[*]The work was done while the author was an intern at Microsoft Research and a graduate student at Stanford.

36th Conference on Neural Information Processing Systems (NeurIPS 2022).

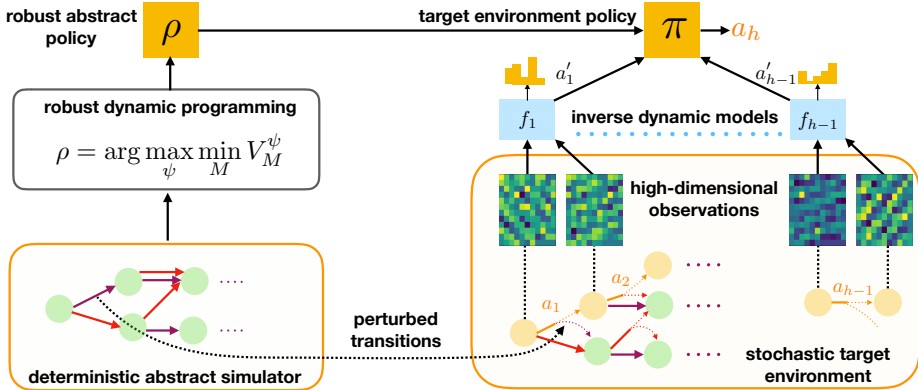

Figure 1: Overview of our setup. We propose an algorithm TASID that learns a robust policy $\pi$ in the target environment using side information in the form of a fully specified deterministic model, called an "abstract simulator." Abstract simulator approximates (possibly stochastic) latent dynamics of the target environment and thus serves as an idealized description of the the environment. The policy $\pi$ selects its actions according to the robust abstract policy $\rho$, but the latent states need to be inferred from high-dimensional observations using learned inverse dynamics models.

not have exactly their intended effect. More complex simulators might be more faithful, but simpler simulators might be easier to build and also more computationally tractable.

In this paper, we address this trade-off with a particular design choice: First, we assume that the abstract simulator is deterministic. Indeed, an ordinary map, which implicitly represents what new position will be reached by a particular action, is such a deterministic model. Compared to fully probabilistic models, deterministic ones are especially simple to build and work with.

On the other hand, a deterministic model will not in general capture the "noise" of real-world dynamics. Therefore, we seek algorithms that will learn policies that are robust to errors or "perturbations" in the model represented by the abstract simulator. To this end, in Section 4, we present a new algorithm called TASID that provably finds a policy achieving a certain level of performance for any target environment whose dynamics can be reasonably approximated by a given abstract simulator, up to some quantifiable perturbation level, and with arbitrarily different observations. (Figure 1 visualizes our setup and approach.)

In the robot example, although a map is intuitively helpful for reducing the need for exploration, there is still much to be inferred, for instance, how locations on the map correspond to locations in the building, and, even more challenging, how they correspond to rich, high-dimensional observations experienced via cameras or other sensors. Our algorithm solves these challenges, building on prior work on learning from such observations [Krishnamurthy et al., 2016, Jiang et al., 2017].

Furthermore, our theoretical guarantees show that we do indeed save dramatically on sample complexity (measured by the required number of interactions with the target environment) by not having to explore the entire environment, and instead focusing the task of learning observations just to the states that lead towards accomplishing the task. Indeed, our algorithm's sample complexity is entirely *independent* of the size of the state space, assuring efficiency even when the state space is extremely large, as is often the case. We are able to achieve this result because of specific assumptions and criteria for success, as outlined above, namely, near-deterministic latent dynamics, knowledge of the abstract simulator, and the benchmark of a robust policy rather than an optimal policy.

In Section 5, we empirically evaluate TASID in two domains. The first represents a challenging problem requiring strategic exploration. We show that TASID is able to efficiently achieve the robust policy value while the PPO algorithm [Schulman et al., 2017] augmented with exploration bonus [Burda et al., 2019b] and domain randomization [Tobin et al., 2017] does not solve the problem. The second domain is a visual grid world that tests the robustness of TASID to dynamics perturbation and the change of state-space size. We show that TASID succeeds empirically in agreement with our theory.

## 2 Related Work

**Transfer RL.** The goal of transfer RL is to speed up learning in a target environment by leveraging a model (or some other form of knowledge) learnt in a source environment [Taylor and Stone, 2009, Lazaric, 2012]. For our setting, the most relevant are the methods that can transfer between environments with different observation spaces. The dominant line of research requires access to a hand-coded inter-task mapping, which is used to introduce constraints, shape rewards, or warm-start learning in the target environment [Taylor and Stone, 2005, Taylor et al., 2007a, Fernández et al., 2010, Mann and Choe, 2013, Brys et al., 2015]. Instead of hand-coding, several works seek to learn inter-task mappings using structural information such as factorization [Liu and Stone, 2006, Taylor et al., 2007b, Soni and Singh, 2006]. Van Driessel and Francois-Lavet [2021] and Sun et al. [2022] do not require such structural information, and only posit assumptions similar to our block MDP assumption, but without requiring near-deterministic latent dynamics. Latent dynamics learnt in the source domain is used to create auxiliary tasks that regularize learning in the target domain.

These methods show empirical promise of transferring information about latent dynamics. Their focus is mainly on modeling and experiments, whereas our work also seeks to derive theoretical guarantees that showcase the benefits of transfer. We derive sample complexity that is independent of the size of the state space and observation space, which would be impossible without transfer. Among the cited works, Mann and Choe [2013] also show benefits of transfer, but their sample complexity generally scales linearly with the size of the target *observation* space. The strong bound that we derive requires a near-deterministic latent dynamics, and so it is not as broadly applicable.

**Provably efficient RL with rich observations.** Our modeling setup and theoretical analysis build on the prior work that considers large-dimensional "rich" observations, but simpler (tabular) latent dynamics [Krishnamurthy et al., 2016, Jiang et al., 2017, Du et al., 2019]. To circumvent the hardness of learning in general partially-observed MDPs (POMDPs), these methods impose structural assumptions, such as the block MDP assumption used here. These methods, like ours, learn with sample complexity that is independent of the size of the observation space, but, unlike ours, that is polynomial in the size of the latent state space. Using existing MDP lower bounds [Kakade, 2003], this dependence is unavoidable for general latent dynamics, even when the decoding function of observations to latent states is known. We overcome this fundamental obstacle by assuming that the latent dynamics is near-deterministic, and the deterministic approximation is known.

**Robust RL.** Our solution concept and the underlying algorithm build on robust dynamic programming methods [Iyengar, 2005, Bagnell et al., 2001]. However, we are using this concept in a fundamentally new way to achieve sample complexity improvements in transfer learning. This is one of the key conceptual innovations of our work.

## 3 Problem Setting

We next formalize our modeling setup and assumptions. As a running example, we consider a navigation problem on a floor of a building, where the goal is to reach a certain location by a robot that can turn left and right and move forward. The robot senses the environment via lidar readings and/or a camera, but it does not have access to its position or orientation. We use the notation $\Delta(S)$ for the set of probability distributions over a set $S$, and $[n]$ to denote the set $\{1, 2, \ldots, n\}$.

### 3.1 Target environment

We assume that the target environment is a *block MDP* (see, e.g., Du et al., 2019, Misra et al., 2020, Zhang et al., 2020). In a block MDP, observations are high-dimensional (e.g., lidar and camera readings), but emitted by a finite number of latent states (e.g., location and orientation of a robot); latent states are not directly observed, but they can be determined from the observations they emit.

Formally, a block MDP is a triple $\mathbf{M} = \langle M, \mathcal{X}, q \rangle$. The first component of the triple is a standard episodic MDP $M = \langle \mathcal{S}, \mathcal{A}, s_{\text{init}}, H, T, R \rangle$, referred to as the *latent MDP*, with a finite state space $\mathcal{S}$, referred to as the *latent state space*, a finite action space $\mathcal{A}$, initial latent state $s_{\text{init}}$, horizon $H$, transition function $T : [H] \times \mathcal{S} \times \mathcal{A} \to \Delta(\mathcal{S})$, and reward function $R : [H] \times \mathcal{S} \times \mathcal{A} \to [0, 1]$; transition probabilities and reward function values are written as $T_h(s' \mid s, a)$ and $R_h(s, a)$. The second component of the block MDP triple is an *observation space* $\mathcal{X}$, which is typically large and

possibly infinite. The final component is an *emission function* $q : \mathcal{S} \to \Delta(\mathcal{X})$, which describes the conditional distribution over observations given any latent state; its values are written as $q(s \mid x)$.

We assume that $q$ satisfies the block MDP assumption, meaning that there exist disjoint sets $\{\mathcal{X}_s\}_{s \in \mathcal{S}}$ such that if $x \sim q(\cdot \mid s)$ then $x \in \mathcal{X}_s$ with probability 1. This means that there exists a *perfect decoder* $\phi_{\mathbf{M}} : \mathcal{X} \to \mathcal{S}$ that maps an observation $x$ to the unique state $\phi_{\mathbf{M}}(x)$ that emits it.

An agent interacts with a block MDP in a sequence of episodes, each generated as follows: Initially, $s_1 = s_{\text{init}}$. At each step $h = 1, \ldots, H$, the agent observes $x_h \sim q(\cdot \mid s_h)$, then takes action $a_h$, accrues reward $r_h = R_h(s_h, a_h)$, after which the MDP transitions to state $s_{h+1} \sim T_h(\cdot \mid s_h, a_h)$. The agent does *not* observe the latent states $s_h$, only the observations $x_h$ and rewards $r_h$. Observations and actions up to step $h$ are denoted $\mathbf{x}_{1:h}$ and $\mathbf{a}_{1:h}$.

We denote the block MDP describing the target environment by $\mathbf{M}^\star = \langle M^\star, \mathcal{X}, q^\star \rangle$ where $M^\star = \langle \mathcal{S}, \mathcal{A}, s_{\text{init}}, H, T^\star, R^\star \rangle$ is the target latent MDP. The perfect decoder for $\mathbf{M}^\star$ is denoted as $\phi^\star$.

**Practicable policy.** Behavior of an agent in a target environment is formalized as a (non-Markovian) *practicable policy*. A practicable policy prescribes which action to take given any sequence of observations, i.e., it is a mapping $\pi : \mathcal{X}^{\leq H} \to \mathcal{A}$, where $\mathcal{X}^{\leq H} = \cup_{h=1}^{H} \mathcal{X}^h$. The action taken by $\pi$ on $\mathbf{x}_{1:h}$ is written $\pi_h(\mathbf{x}_{1:h})$. The expected sum of rewards in $\mathbf{M}$, when actions $a_h$ are chosen according to a practicable policy $\pi$, i.e., when $a_h = \pi_h(\mathbf{x}_{1:h})$, is referred to as the *value of $\pi$ in $\mathbf{M}$* and denoted $V_{\mathbf{M}}^\pi = \mathrm{E}_{\mathbf{M},\pi}[r_1 + r_2 + \cdots + r_H]$; the subscript in the expectation signifies the probability distribution over episode realizations when the environment follows $\mathbf{M}$, and actions are chosen according to $\pi$.

We also allow practicable policies to be (Markovian) mappings $\pi : [H] \times \mathcal{X} \to \mathcal{A}$ which choose every action according to the last observation, so that $a_h = \pi_h(x_h)$ for all $h$. Whether a particular practicable policy is Markovian or not will generally be clear from context.

## 3.2 Abstract simulator

Our learning algorithm can interact with the target environment, but it additionally has access to an *abstract simulator*, which provides an idealized and abstracted version of the target environment. Formally, an abstract simulator is an episodic MDP denoted $M^\circ = \langle \mathcal{S}, \mathcal{A}, s_{\text{init}}, H, T^\circ, R^\circ \rangle$, with the same state space, action space, start state and horizon as the latent MDP $M^\star$, but not necessarily the same transition and reward functions. Furthermore, we assume that abstract simulator is deterministic:

**Assumption 1.** *The abstract simulator $M^\circ$ is deterministic, i.e., $T_h^\circ(s_{h+1} \mid s_h, a_h) \in \{0, 1\}$.*

In order for the abstract simulator to be useful, it must approximate target environment. In this paper we assume that the target environment can be viewed as a "perturbed" version of the abstract simulator, using a notion of perturbation inspired by the concept of trembling-hand equilibria from extensive-form games [Selten, 1975]. Specifically, we say that an MDP $M'$ is an $\eta$-perturbation of another MDP $M$ if its dynamics can be realized by following $M$'s dynamics while distorting agent actions according to some (unknown) "noise" distribution, referred to as $\xi$ in the definition below, which keeps actions unchanged with probability at least $1 - \eta$:

**Definition 1** ($\eta$-perturbation)**.** *We say that an MDP $M' = \langle \mathcal{S}, \mathcal{A}, s_{init}, H, T', R' \rangle$ is an $\eta$-perturbation of an MDP $M = \langle \mathcal{S}, \mathcal{A}, s_{init}, H, T, R \rangle$ if there exists a function $\xi : [H] \times \mathcal{S} \times \mathcal{A} \to \Delta(\mathcal{A})$ that satisfies $\xi_h(a \mid s, a) \geq 1 - \eta$ for all $h$, $s$, $a$, and such that*

$$T_h'(s' \mid s, a) = \sum_{a' \in \mathcal{A}} T_h(s' \mid s, a') \xi_h(a' \mid s, a)$$
$$R_h'(s, a) = \sum_{a' \in \mathcal{A}} R_h(s, a') \xi_h(a' \mid s, a)$$

*for all $h$, $s$, $a$, $s'$; thus MDP $M'$ can be viewed as following the dynamics of $M$ in which each action $a$ is stochastically replaced ("perturbed") according to $\xi$.*

*The set of all $\eta$-perturbations of $M$ is denoted $\mathcal{C}(M, \eta)$.*

We assume that the target environment is an $\eta$-perturbation of the abstract simulator for a value of $\eta < 0.5$. Thus, most of the time the target environment transitions after each action "as intended" (i.e., following known dynamics of the abstract simulator), but with a probability at most $\eta$ it may depart from the intended action due to an inherent, but unknown stochasticity:

**Assumption 2.** *$M^\star$ is an $\eta$-perturbation of the abstract simulator $M^\circ$ for some $\eta < 0.5$.*

**Abstract policy.** Behavior of an idealized agent that can directly access latent state is formalized as a (Markovian) *abstract policy*. An abstract policy prescribes what action to take in each state $s$ at a given step $h$, i.e., it is a mapping $\psi : [H] \times \mathcal{S} \to \mathcal{A}$; we write $\psi_h(s)$ for the action taken by $\psi$ in step $h$ and state $s$. The expected sum of rewards in an episodic MDP $M = \langle \mathcal{S}, \mathcal{A}, s_{\text{init}}, H, T, R \rangle$, when following $\psi$ is called the *value of $\psi$ in $M$* and denoted $V_M^\psi = \mathrm{E}_{M,\psi}[r_1 + r_2 + \cdots + r_H]$.

**Robust abstract policy.** Since the latent MDP in the target domain is a perturbation of the abstract simulator, we will seek to obtain policies that are robust to *any* allowed perturbation. For a given abstract simulator $M^\circ$ and the perturbation level $\eta$, we define a *robust abstract policy* $\rho$ to be a policy that achieves the largest possible value under the worst-case choice of perturbation:

$$\rho = \underset{\psi \in \Psi}{\mathrm{argmax}} \min_{M \in \mathcal{C}(M^\circ, \eta)} V_M^\psi, \tag{1}$$

where $\Psi$ is the set of all mappings from $[H] \times \mathcal{S}$ to $\mathcal{A}$.

In a natural way, a robust abstract policy $\rho$ can be composed with the perfect decoder $\phi^\star$ to obtain a (Markovian) practicable policy $\rho \circ \phi^\star : [H] \times \mathcal{X} \to \mathcal{A}$ mapping observations in the target environment to actions while still maximizing the worst-case reward among perturbations of $M^\circ$. We aim for algorithms that find practicable policies that perform almost as well as this *robust practicable policy*.

### 3.3 The learning setting

We can now formally define our learning setting. A learning algorithm `Alg` in this setting receives as input a deterministic abstract simulator (episodic MDP) $M^\circ = \langle \mathcal{S}, \mathcal{A}, s_{\text{init}}, H, T^\circ, R^\circ \rangle$, meaning it receives the entire MDP represented in tabular form (or some other computationally convenient form). The algorithm is also provided with *oracle access* to a target environment (block MDP), $\mathbf{M}^\star = \langle M^\star, \mathcal{X}, q^\star \rangle$. This means that the algorithm cannot directly access $\mathbf{M}^\star$ itself, but can interact with it as an agent would, executing actions $a_h$, and receiving back observations $x_h$ and rewards $r_h$ in a sequence of episodes, as described above. Finally, `Alg` is given parameters $\epsilon > 0$, $\delta > 0$, and $\eta < 0.5$. It is assumed that $M^\star$ is an $\eta$-perturbation of $M^\circ$. The assumption that we know the value of $\eta$ is not essential and we discuss in Section 4.4 how it can be removed.

After interacting with $\mathbf{M}^\star$, the algorithm outputs a practicable policy $\pi$. The goal of learning is for $\pi$ to have value almost as good as the robust practicable policy with high probability, that is, for $V_{\mathbf{M}^\star}^\pi \geq V_{\mathbf{M}^\star}^{\rho \circ \phi^\star} - \epsilon$ with probability at least $1 - \delta$ (where probability is over the algorithm's randomization as well as randomness in the interactions with the target environment). Furthermore, we require the number of episodes executed by the algorithm before outputting a policy $\pi$ to be bounded by a polynomial in the number of actions $|\mathcal{A}|$, the horizon $H$, $1/\epsilon$, $1/\delta$, and $1/(1 - 2\eta)$. Note importantly that this polynomial must have no explicit dependence on the number of states $|\mathcal{S}|$ or observations $|\mathcal{X}|$ in the target environment. An algorithm that satisfies these criteria (given the stated assumptions) is said to achieve an *efficient transfer from abstract simulator*.

## 4 Main Algorithm

Our main contribution is an algorithm `TASID`, which achieves efficient transfer from an abstract simulator. The algorithm operates in two stages. In the first stage, it determines the robust abstract policy $\rho$ for the provided abstract simulator via robust dynamic programming (Algorithm 1). In the second stage, it interacts with the target environment (via oracle access) in order to learn to predict the current latent state based on the history of observations and actions. The learnt decoding map is then composed with the robust abstract policy to obtain the practicable policy that is returned by the algorithm (see Algorithm 2).

### 4.1 Robust dynamic programming for abstract simulator

We obtain the robust abstract policy by instatiating the robust dynamic programming algorithm of Bagnell et al. [2001] and Iyengar [2005] to our specific notion of perturbation. The algorithm proceeds by filling out values of the *robust value function* $\tilde{V}$, which quantifies the largest sum of rewards achievable starting at any step $h$ and state $s$, when assuming the worst-case perturbation of the input MDP $M^\circ$. Specifically,

$$\tilde{V}_h(s) = \max_{\psi \in \Psi} \min_{M \in \mathcal{C}(M^\circ, \eta)} \mathrm{E}_{M,\psi}\big[r_h + r_{h+1} + \cdots + r_H \mid s_h = s\big].$$

---

**Algorithm 1** Robust Dynamic Programming. $\text{RDP}(M^\circ, \eta)$

---

**Input:** An episodic MDP $M^\circ = \langle \mathcal{S}, \mathcal{A}, s_{\text{init}}, H, T^\circ, R^\circ \rangle$, perturbation level $\eta$.

1: $\tilde{V}_{H+1}(s) \leftarrow 0$ for all $s \in \mathcal{S}$
2: **for** $h = H, \ldots, 1$ **do**
3:      **for** all $s \in \mathcal{S}, a \in \mathcal{A}$:    $\tilde{Q}_h(s,a) \leftarrow R_h^\circ(s,a) + \sum_{s'} T_h^\circ(s' \mid s, a) \tilde{V}_{h+1}(s')$
4:      **for** all $s \in \mathcal{S}$:           $\tilde{V}_h(s) \leftarrow (1-\eta) \max_a \tilde{Q}_h(s,a) + \eta \min_a \tilde{Q}_h(s,a)$
5: **Return** $\rho$ defined by $\rho_h(s) = \text{argmax}_a \tilde{Q}_h(s,a)$

---

---

**Algorithm 2** Transfer from Abstract Simulator using Inverse Dynamics. $\text{TASID}(\mathbf{M}^\star, M^\circ, \mathcal{F}, \eta, \epsilon, \delta)$

---

**Input:** Oracle access to target environment $\mathbf{M}^\star$, deterministic abstract simulator $M^\circ$,
         optimization-oracle access to a function class $\mathcal{F} \subseteq \{\mathcal{X}^2 \to \Delta(\mathcal{A})\}$,
         perturbation level $\eta < 0.5$, target accuracy $\epsilon > 0$, failure probability $\delta$.

1: Let $\rho$ be the robust policy returned by $\text{RDP}(M^\circ, \eta)$
2: Let $n_D := \frac{8H^2 |\mathcal{A}|^3 \ln(|\mathcal{F}|/\delta)}{\epsilon(1-2\eta)^2}$
3: Let $\pi_1(x) := \rho_1(s_{\text{init}})$ for all $x \in \mathcal{X}$               // Define practicable policy in step $h = 1$
4: **for** $h = 1, \ldots, H-1$ **do**
5:      $\mathcal{D}_h \leftarrow \emptyset$               // Gather dataset $\mathcal{D}_h$ for learning "inverse dynamics" in step $h$
6:      **for** $n_D$ times **do**
7:          Follow $\pi_{1:h-1}$ for $h-1$ steps to observe $x_h$
8:          Take action $a_h$ uniformly at random and observe $x_{h+1}$
9:          $\mathcal{D}_h \leftarrow \mathcal{D}_h \cup \{(x_h, a_h, x_{h+1})\}$
10:      $f_h := \arg\max_{f \in \mathcal{F}} \sum_{(x_h, a_h, x_{h+1}) \in \mathcal{D}_h} \ln f(a_h \mid x_h, x_{h+1})$      // Learn "inverse dynamics"
11:      Define $\alpha_h : \mathcal{X}^2 \to \mathcal{A}$ as               // Define "shadow action" decoder
         $\alpha_h(x_h, x_{h+1}) = \text{argmax}_{a \in \mathcal{A}} f_h(\cdot \mid x_h, x_{h+1})$
12:      Define $\phi_{h+1} : \mathcal{X}^{h+1} \to \mathcal{S}$ such that               // Define state decoder
         $\phi_{h+1}(\mathbf{x}_{1:h+1})$ is the state $s_{h+1}$ reached in $M^\circ$, when starting in $s_{\text{init}}$ and executing
           $a_1' = \alpha_1(x_1, x_2), \ldots, a_h' = \alpha_h(x_h, x_{h+1})$
13:      Define $\pi_{h+1} : \mathcal{X}^{h+1} \to \mathcal{A}$ as               // Define practicable policy
         $\pi_{h+1}(\mathbf{x}_{1:h+1}) = \rho_{h+1}\big(\phi_{h+1}(\mathbf{x}_{1:h+1})\big)$
14: **return** $\pi = (\pi_1, \ldots, \pi_H)$

---

Similar to standard dynamic programming, the value function in robust dynamic programming can be filled out beginning with $h = H + 1$, where we have $\tilde{V}_h(s) = 0$, and proceeding backward. In our case, the values $\tilde{V}_h(s)$ can be obtained from $\tilde{V}_{h+1}(s)$ using a closed-form expression (line 4), which leverages intermediate values $\tilde{Q}_h(s,a)$. Note that $\tilde{Q}$ is not quite the robust state-action value function, because it assumes that the action at step $h$ is left unperturbed (and only considers the worst-case perturbation in the following steps). The function $\tilde{Q}$ is used to derive the robust policy (line 5).

**Theorem 1.** *For any abstract simulator $M^\circ$ and any perturbation level $\eta$, Algorithm 1 returns the robust policy $\rho = \text{argmax}_{\psi \in \Psi} \min_{M \in \mathcal{C}(M^\circ, \eta)} V_M^\psi$.*

(The proof of this theorem and all other proofs in this paper are deferred to the appendix.)

## 4.2    Learning a decoder in the target environment

We cannot directly apply the robust abstract policy in the target environment, because the latent states are not observable. Therefore, we construct a "state decoder" in the target environment, which uses the history of observation in an episode to predict the current latent state; the state decoder is combined with the robust abstract policy to obtain the practicable policy. Formally, a (non-Markovian) state decoder is a mapping $\phi : \mathcal{X}^{\leq H} \to \mathcal{S}$. In step $h$ of an episode, the history of previous observations $\mathbf{x}_{1:h}$ is used as an input, and the decoder predicts the latent state $s_h$. To construct such a state decoder, we crucially leverage the abstract simulator $M^\circ$.

The abstract simulator $M^\circ$ is deterministic, so a specific sequence of actions $a_1, \ldots, a_h$ always leads to the same state. The latent MDP in the target environment is a perturbation of the abstract simulator. This means that when an agent takes action $a_h$ in step $h$, most of the time the environment transitions according to $T^\circ(\cdot \mid s_h, a_h)$, but sometimes (with probability at most $\eta$) it transitions according to some other action. The action $a_h$ gets replaced with some "shadow" action $a'_h$ according to an unknown noise distribution $\xi$, and the latent state then transitions according to $T^\circ(\cdot \mid s_h, a'_h)$. If we knew shadow actions $a'_1, \ldots, a'_h$, we could then recover the current latent state by simulating that same sequence of actions in the abstract simulator.

To obtain shadow actions, we learn an "inverse dynamics" model, which predicts $a'_h$ from the observations $x_h$ and $x_{h+1}$ (an approach also used in previous work on block MDPs). We learn a separate inverse dynamics model for each step of an episode. In step $h$, we sample triplets of the form $(x_h, a_h, x'_h)$ across multiple episodes in the target environment, and then fit a model $f_h$ for the conditional probability of $a_h$ given $x_h$ and $x'_h$. The model $f_h(a_h \mid x_h, x'_h)$ is referred to as an inverse dynamics model, because it "inverts" the dynamics represented by the transition function. We show that if we were able to obtain an exact model $f_h^\star$ of the conditional probability, then the action $a$ with the largest probability $f_h^\star(a \mid x_h, x'_h)$ would be the correct shadow action.

As is standard in the block MDP literature, in order to fit an inverse dynamics model, we assume access to an optimization algorithm capable of fitting functions from some class $\mathcal{F} \subseteq \{\mathcal{X}^2 \to \Delta(\mathcal{A})\}$ to data; we call this algorithm an *optimization oracle for $\mathcal{F}$*. The class $\mathcal{F}$ should be sufficiently expressive to approximate the required conditional probability distribution.

We now have all the pieces required to describe our algorithm. The algorithm first constructs the robust abstract policy (line 1), and defines the practicable policy $\pi_1$ at the initial step $h = 1$, where the latent state is known to be $s_{\text{init}}$ (line 3). The algorithm then proceeds iteratively to fill in $\pi_2, \ldots, \pi_H$. In iteration $h$, the algorithm first learns the inverse dynamics model $f_h$ with the help of the optimization oracle (line 5–10). The inverse dynamics model is then used to obtain the shadow action decoder $\alpha_h$ (line 11), which predicts which action caused the transition from $x_h$ to $x_{h+1}$. Using the shadow action decoders up to step $h$, we can construct a state decoder $\phi_{h+1}$ (line 12), which for a given history of observations $\mathbf{x}_{1:h+1}$, first predicts their corresponding shadow actions $a'_1, \ldots, a'_h$ and then uses the abstract simulator to determine the state $s_{h+1}$ that they lead to. Finally, using the state decoder $\phi_{h+1}$, we define the practicable policy at the step $h + 1$ to return the same action as the abstract robust policy would return on the decoded state (line 13).

### 4.3 Sample Complexity of TASID

We next provide the sample complexity analysis of TASID, showing that it indeed achieves efficient transfer from an abstract simulator. In addition to Assumptions 1 and 2, we also need to ensure that the function class $\mathcal{F}$ is expressive enough to approximate the conditional probability distribution being fitted by the inverse dynamics model. It turns out that this target probability distribution can be expressed in terms of the transition function of the block MDP $\mathbf{M}^\star$, which is the function $\mathbf{T}^\star : [H] \times \mathcal{X} \times \mathcal{A} \to \Delta(\mathcal{X})$ equal to

$$\mathbf{T}_h^\star(x' \mid x, a) := q^\star(x' \mid s' = \phi^\star(x')) T_h\big(s' = \phi^\star(x') \mid s = \phi^\star(x), a\big).$$

Using $\mathbf{T}^\star$, we can write the exact inverse dynamics model as $f_h^\star(a \mid x, x') = \frac{\mathbf{T}_h^\star(x'\mid x,a)}{\sum_{a'\in\mathcal{A}} \mathbf{T}_h^\star(x'\mid x,a')}$.

To state our assumption, let $P_h$ denote the distribution over triples $(x_h, a_h, x_{h+1})$ sampled in line 7-8, and let $\widetilde{f}_h$ be the maximizer (in $\mathcal{F}$) of the expected log likelihood under $P_h$:

$$\widetilde{f}_h := \underset{f \in \mathcal{F}}{\arg\max}\, \mathrm{E}_{x,a,x'\sim P_h}[\ln f(a \mid x, x')].$$

**Assumption 3** (Approximate realizability). *There exists $\epsilon_\mathcal{F} \geq 0$ such that for every $h \in [H]$, and for all triples $(x, a, x')$ in the support of $P_h$*

$$f_h^\star(a \mid x, x') \leq (1 + \epsilon_\mathcal{F})\widetilde{f}_h(a \mid x, x').$$

Exact realizability assumptions (with $\epsilon_\mathcal{F} = 0$) are standard in the block MDP literature [Du et al., 2019, Agarwal et al., 2020]. Here we only assume approximate realizability, which can be assured for any $\epsilon_\mathcal{F} > 0$ by choosing a sufficiently expressive class such as deep neural networks. In practice, we choose classes that express some inductive biases (such as feature invariance, factorization, etc.).

We are now ready to state our main theoretical result—sample complexity of TASID.

**Theorem 2.** *Let $M^\circ$ be a deterministic abstract simulator. Let $\mathbf{M}^\star$ be a target environment for which $\mathbf{M}^\star$ is an $\eta$-perturbation of $M^\circ$ for some $\eta < 0.5$. Let $\mathcal{F}$ be a class of functions that satisfies Assumption 3 with $\epsilon_\mathcal{F} \leq \frac{(1-2\eta)^2}{8H^4|\mathcal{A}|^3}$. Then for any $\epsilon > 0$ and $\delta \in (0,1)$, Algorithm 2 with oracle access to $\mathbf{M}^\star$, optimization-oracle access to $\mathcal{F}$, and inputs $M^\circ$, $\eta$, $\epsilon/2$, and $\delta$, executes $n = \mathcal{O}\big(\frac{H^3|\mathcal{A}|^3 \ln(|\mathcal{F}|/\delta)}{\epsilon(1-2\eta)^2}\big)$ episodes and returns a practicable policy $\pi$ that with probability at least $1 - \delta$ satisfies*

$$V_{\mathbf{M}^\star}^\pi \geq V_{\mathbf{M}^\star}^{\rho \circ \phi^\star} - \epsilon.$$

Thus, Algorithm 2 achieves efficient transfer from an abstract simulator, meaning that its sample complexity is independent of the sizes of $|\mathcal{S}|$ and $|\mathcal{X}|$. We also achieve a fast rate, $\mathcal{O}(1/\epsilon)$, with respect to the sub-optimality $\epsilon$. The assumption on $\epsilon_\mathcal{F}$ is not restrictive, because we can always choose a sufficiently expressive class. Note that previous works simply assume $\epsilon_\mathcal{F} = 0$ [e.g., Du et al., 2019, Agarwal et al., 2020, Misra et al., 2020]. The dependence on $\ln |\mathcal{F}|$ is standard; for neural networks it is proportional to the number of bits to store the network weights. For example, if the neural network has 1000 weights represented by doubles (i.e., with 64 bits) then $\ln |\mathcal{F}| = (\ln 2) \cdot 64000$. There may be room to improve dependence on the horizon $H$ and action space size $|\mathcal{A}|$, but these were not our focus here.

### 4.4 Extensions

In the appendix, we show how `TASID`, under additional assumptions, can be extended to settings where the abstract simulator has a stochastic start state (but deterministic transitions).

The assumption that $\eta$ is known is not necessary. If $\eta$ is not known, it is possible to carry out a form of binary search as follows: Run the algorithm with $\eta = 1/4$ and verify whether it achieves the performance guarantee. If it does, continue with $\eta = 1/8$ (splitting the interval $(0, 1/4]$), otherwise continue with $\eta = 3/8$ (splitting the interval $(1/4, 1/2)$), and run the algorithm again. If the goal is to recover $\eta$ up to the precision $\epsilon_\eta$, then we just need to repeat this $\log_2(1/2\epsilon_\eta)$ times.

Although our focus here is on transfer from an abstract simulator, and we emphasize the relative ease of constructing abstract simulators, our approach could also be used for transfer learning between two block MDP environments that share latent space structure. In the source domain, we could use an existing block MDP approach [e.g., Misra et al., 2020] to infer latent structure, which could then be used as an abstract simulator to speed up learning in the target domain using `TASID`.

## 5 Experiments

We evaluate `TASID` in two simulation environments, focusing on four of its properties: sample complexity independent of state-space size; scalability to complex visual observations; robustness of the policy it learns; and scalability to large state spaces. We summarize the results here and defer the details, including hyperparameter selection, detailed environment descriptions, algorithm and baselines implementation and a full description of results under various setups to the appendix.

**Can `TASID` solve problems that require strategic exploration?** We theoretically showed that `TASID` can solve problems that require performing strategic exploration using a small number of episodes. We test this empirically on a challenging environment called *combination lock*.

**Combination lock.** We first describe the abstract simulator $M^\circ$ visualized in Figure 2a. It has an action space $\mathcal{A}$, horizon $H$, and a state space $\{(h, i) \mid h \in \{0\} \cup [H], i \in [3]\}$, with the initial state $(0, 1)$. As we will see, states $\{(h, 1), (h, 2) \mid h \in [H]\}$ are *good* states from which optimal return is possible, while states $\{(h, 3) \mid h \in [H]\}$ are *bad* states. In state $(h, 1)$, one good action leads to state $(h + 1, 1)$, and all other $|\mathcal{A}| - 1$ actions lead to state $(h + 1, 2)$. In state $(h, 2)$, one good action leads to state $(h + 1, 2)$, another good action leads to state $(h + 1, 1)$, and all other $|\mathcal{A}| - 2$ actions lead to $(h + 1, 3)$. All actions in state $(h, 3)$ lead to state $(h + 1, 3)$. The identities of good actions are unknown and are different for different states. Any transition to state $(H, 1)$ gives a reward of $9.5$, whereas transition to state $(H, 2)$ gives a reward of $10$, and transition to state $(H, 3)$ gives a reward of $0$. To "mislead" the agent, transitions from $(h, 2)$ to $(h + 1, 3)$ give a reward of $1$ and transitions from $(h, 1)$ to $(h + 1, 1)$ give a reward of $-1/H$. All other transitions give a reward of $0$.

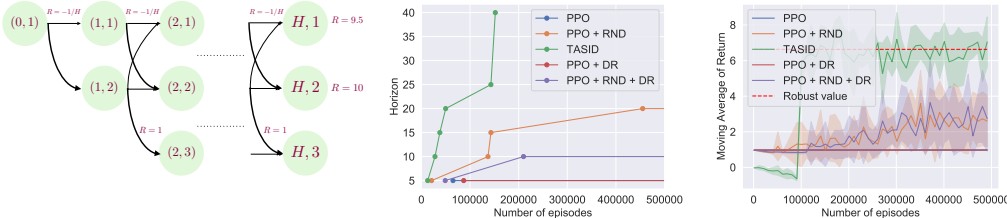

| (a) Combination lock environment | (b) Horizon solved *vs* episodes | (c) Reward curve for $H = 40$ |

Figure 2: (a): Latent MDP in the combination lock. (b): The largest problem (expressed as horizon $H$) that each algorithm can solve (defined as reaching 95% of the value of the robust policy) within $500\,000$ episodes; we consider $H = 5, 10, 15, 20, 25, 40$ and $|\mathcal{A}| = 10$. We report median number of episodes across five trials with different seeds (see the appendix for full details). (c): Total reward per episode for the problem size $H = 40$. TASID is a batch algorithm trained on the first $100\,000$ episodes, thus its performance before the end of training is very low and after the end of training stays constant. Shaded bands correspond to the $95\%$ Student's $t$ confidence intervals (here and elsewhere in paper).

The latent MDP $M^\star$ of the target environment is constructed by taking a random $\eta$-perturbation of $M^\circ$ for $\eta = 0.1$. Exploration in both $M^\star$ and $M^\circ$ can be difficult due to misleading rewards and challenging dynamics, where most actions lead to bad states. The optimal policy in $M^\circ$ finishes in the state $(H, 2)$ and achieves the value of 10. However, the robust policy will attempt to visit the state $(H, 1)$ which lies on a more stable trajectory. Meanwhile, the optimal policy in $M^\circ$ will fail in $M^\star$, because perturbations are likely to move the agent into a bad state.

Observations in the target environment are real-valued vectors of dimension $2^{\lceil \log_2(H+4) \rceil}$. For a latent state $(h, i)$, the observation is generated by first creating an $(H + 4)$-dimensional vector by concatenating one-hot encodings of $h$ and $i$, element-wise adding Gaussian noise, applying a fixed coordinate permutation, and finally multiplying the vector with a Hadamard matrix.

We compare TASID with four baselines. The first is PPO [Schulman et al., 2017], a policy-gradient-based algorithm. The second is PPO-RND, which augments PPO with an exploration bonus based on prediction error [Burda et al., 2019b]. These two baselines do not use abstract simulator and run on the target environment from scratch. The other two baselines, PPO+DR and PPO+RND+DR, enhance PPO and PPO+RND with domain randomization (DR), where the policy is pre-trained on a set of randomized block MDPs [Tobin et al., 2017]. PPO+DR and PPO+RND+DR are designed for transfer RL, but unlike our work, they rely on observation-based similarity. During the pre-training phase of domain randomization, we generate block MDPs by following a similar process as the one used to generate the target environment. However, when generating the emission function, we only generate a random permutation matrix and the other components are kept the same as in the target environment. This randomization set includes the true emission function and transition dynamics.

For baseline algorithms, we run grid search over hyperparameters listed in Table 3 in Appendix D, separately for each environment specification (each value of $H$), and report the best results of PPO(+RND)(+DR). For TASID, we consider only one hyperparameter, the number of training episodes per time step $n_D$, and search over three possible values: 1000, 2500, 10000.

Figure 2b plots the size of the problem (expressed as horizon $H$) that each algorithm can solve (defined as reaching at least 95% of the value of the robust policy) within a given number of episodes; we considered $H = 5, 10, 15, 20, 25, 40$. The plot shows that PPO and PPO+DR fail to solve the problem beyond the smallest size ($H = 5$), which is not surprising as they are not designed to perform strategic exploration. PPO+RND can solve the problem until $H = 20$, but fails for $H = 25$, showing that RND bonus helps to an extent but cannot solve harder problems. Interestingly, PPO+RND+DR underperforms compared to PPO+RND even though it has access to pre-training. We believe this is because RND reward bonus does not work well with domain randomization which can mislead the RND bonus by randomizing the observation. Finally, TASID can solve the problem for all values of $H$ and is more sample efficient than the baselines.

**Can TASID scale to complex observation spaces?** We evaluate TASID in the visual MiniGrid environment [Chevalier-Boisvert et al., 2018] with noisy observations. We test on the grid world

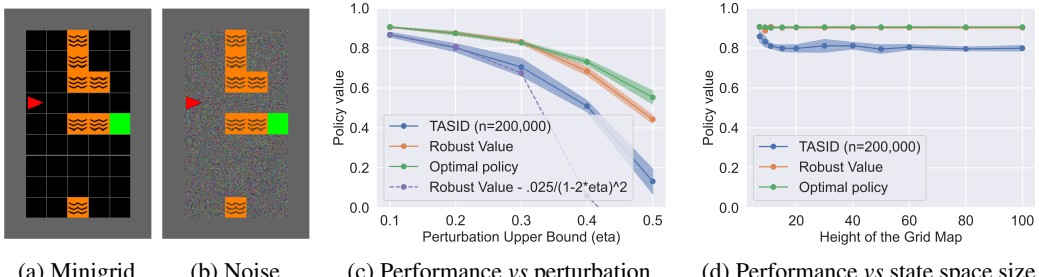

|              |           |                          |                              |
|:------------:|:---------:|:------------------------:|:----------------------------:|
| (a) Minigrid | (b) Noise | (c) Performance *vs* perturbation | (d) Performance *vs* state space size |

Figure 3: (a,b): Map of a $7 \times 11$ MiniGrid environment with noisy observations and five actions: forward, turn left, turn right, turn left + forward, turn right + forward. (c): The performance of `TASID` with different amount of perturbation $\eta$. (d): The performance of `TASID` with different heights of the map.

map shown in Figure 3a. The green square represents the goal and orange squares are lava that should be avoided. The agent's position and direction is shown by the red triangle. The agent gets a reward of $1$ on reaching the goal, $-1$ for reaching the lava, and $-0.01$ for every other step. The state encodes the position and direction of the agent and the current time step. There are 5 actions: move forward, turn left, turn right, and combinations of turning and moving. The abstract simulator $M^\circ$ models the deterministic transitions on the grid. The target environment is an $\eta$-perturbation of $M^\circ$ for $\eta = 0.1$. The agent cannot observe the whole map as in Figure 3a, but only an image of the $7 \times 7$ grids ($56 \times 56$ pixels) in the direction it is facing, with i.i.d. random noise added to all black pixels as shown in Figure 3b. This type of random noise has been studied in the literature and has been shown to pose challenge for RL algorithms [Burda et al., 2019a].

In this grid world, the agent can either use the top route to reach the goal, or a bottom route. The top route is shorter but the agent is more likely to visit the lava due to perturbations. Therefore, the robust policy prefers the bottom route which is longer but more robust.

**How robust is `TASID` to different $\eta$?** Our theoretical analysis suggests that the suboptimality scales as $\mathcal{O}(1/(1 - 2\eta)^2)$. We evaluate this error empirically in the grid world environment for various values of $\eta$ in $[0.1, 0.5]$ and show the results in Figure 3c. The performance matches the theoretical prediction of $V^{\rho \circ \phi^\star} - \mathcal{O}(1/(1 - 2\eta)^2)$ until $\eta = 0.3$, and becomes even better for larger $\eta$. Interestingly, even though the theoretical guarantee is vacuous at $\eta = 0.5$, the algorithm still learns a non-trivial policy with the value greater than that of a random policy. Note that we tune the value of $n_D$ rather than use the value suggested by our theory.

**How does `TASID`'s performance scale with $|\mathcal{S}|$?** We showed theoretically that `TASID`'s sample complexity to learn a near-optimal robust policy is independent of the size of the state space. We empirically test this by varying the height of the grid world environment while keeping the width and the horizon $H$, and overall layout the same. Results are presented in Figure 3d. As height changes from 10 to 100, the size of the state space increases ten times; however, we do not see a significant drop in the performance of `TASID` when using a fixed number of episodes in the target environment.

## 6   Conclusion and Future Work

We presented a new algorithm `TASID` for transfer RL in block MDPs that quickly learns a robust policy in the target environment by leveraging an abstract simulator. We have also proved a sample complexity bound, which does not scale with the size of the state space or the observation space. Finally, we demonstrated theoretical properties by empirical evaluation in two domains.

Our work raises many important questions. Our theoretical analysis depends on the assumption that the abstract simulator is deterministic and the the target environment is its perturbation. Is it possible to relax these requirements and still obtain sample bounds that do not scale with the state space size? One direction might be to leverage "full simulators" (including observations), and incorporate observation similarity between the simulator and the real environment, along with similarity in the latent dynamics. Another important question is how to extend the presented approach to continuous control problems in robotics, which are natural domains for sim-to-real transfer.

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
