| Notation | Definition |
|---|---|
| $[N]$ | For any natural number $N \in \mathbb{N}$, denotes the set $\{1, 2, \cdots, N\}$. |
| $\|\mathcal{U}\|$ | Cardinality of a set $\mathcal{U}$. E.g., $\|[N]\| = N$. |
| $\Delta(\mathcal{U})$ | For any countable set $\mathcal{U}$, denotes the set of all probability distribution over $\mathcal{U}$ |
| $\mathrm{unf}(\mathcal{U})$ | Denotes the uniform distribution over a set $\mathcal{U}$. |
| $\mathcal{S}$ | A finite but potentially very large state space. |
| $\mathcal{A}$ | Agent's finite action space. |
| $H$ | Horizon denoting the number of actions the agent can take in an episode. |
| $T$ | For a given $h \in [H], a \in \mathcal{A}$ and states $s, s' \in \mathcal{S}$, $T_h(s' \mid s, a)$ denotes the probability of transitioning to $s'$ when taking action $a$ in state $s$ at time step $h$. |
| $R$ | For a given $h \in [H], a \in \mathcal{A}$ and state $s \in \mathcal{S}$, $R_h(s, a)$ denotes the reward for taking action $a$ in state $s$ at time step $h$. |
| $s_{\mathrm{init}}$ | A deterministic initial state |
| $M$ | Markov Decision Process (MDP) defined by a tuple $(S, \mathcal{A}, s_{\mathrm{init}}, H, T, R)$. |
| $\mathcal{X}$ | Agent's observation space. Can be potentially infinitely large. |
| $q : \mathcal{S} \to \Delta(\mathcal{X})$ | Emission function where for a given $x \in \mathcal{X}$ and $s \in \mathcal{S}$, $q(x \mid s)$ denotes the probability of observing $x$ when the agent is in latent state $s$ |
| $\mathbf{M}$ | A block MDP defined by the tuple $(M, \mathcal{X}, q)$ where $M$ is referred to as the latent MDP since the agent receives observation instead of state which remains latent. |
| $M^\circ$ | Abstract simulator which is a Markov Decision Process (MDP) given by $(\mathcal{S}, \mathcal{A}, s_{\mathrm{init}}, H, T^\circ, R^\circ)$ and satisfies Assumption 1. |
| $\mathbf{M}^\star$ | Target block MDP defined by $(M^\star, \mathcal{X}, q)$ where the target environment's latent MDP is given by $(\mathcal{S}, \mathcal{A}, s_{\mathrm{init}}, H, T^\star, R^\star)$. The latent MDP $M^\star$ satisfies Assumption 2. |
| $\phi^\star : \mathcal{X} \to \mathcal{S}$ | Perfect decoder of the latent state for $\mathbf{M}^\star$. Agent does not have access to this. |
| $\pi : \mathcal{X}^{\leq H} \to \mathcal{A}$ | Non-Markovian practicable policy. Given an observation sequence $\mathbf{x}_{1:h}$, the policy takes action $\pi_h(\mathbf{x}_{1:h})$. |
| $V_{\mathbf{M}}^\pi$ | Value of a practicable policy $\pi$ in a block MDP $\mathbf{M}$. |
| $\psi : [H] \times \mathcal{S} \to \mathcal{A}$ | A non-stationary Markovian abstract policy. Given state $s$ at time step $h$, the policy takes action $\psi_h(s)$. |
| $\Psi$ | Set of all latent policies, i.e., all mappings of the type $[H] \times \mathcal{S} \to \mathcal{A}$. |
| $V_M^\psi$ | Value of abstract policy $\psi$ in MDP $M$. |
| $\eta$ | Perturbation parameter controlling the amount of stochasticity in the target environment. $\eta = 0$ implies deterministic dynamics. |
| $C(M, \eta)$ | Set of all MDPs which are $\eta$-perturbations of $M$. |
| $\rho$ | Robust policy defined Equation 1, |
| $\epsilon$ | Allowed error in learning the robust practicable policy. |
| $\delta$ | Allowed failure probability in learning the robust practicable policy. |
| $\tilde{V}_h(s)$ | Robust value function. See Section 4.1. |
| $\mathcal{F}$ | A function class containing functions of type $\mathcal{X}^2 \to \Delta(\mathcal{X})$, used by TASID to learn inverse dynamics model. |
| $\alpha_h : \mathcal{X}^2 \to \mathcal{A}$ | Learned "shadow action" decoder in TASID for time step $h$. |
| $\phi_h : \mathcal{X}^h \to \mathcal{S}$ | Learned state decoder in TASID for time step $h$. |
| $\mathcal{D}_h$ | Dataset of transitions collected by TASID at time step $h$. |

Table 1: A list of notations.

In the appendices, we include proofs, extended theoretical results as well as experimental details for all claims in the main paper, under the following structure.

- We provide a list of notations in Table 1.

- Appendix A includes the proof for the analysis of Algorithm 1.

- Appendix B includes the proof for the analysis of Algorithm 2.

- Appendix C includes the algorithm and analysis for the case of stochastic initial states, based on the main algorithm.

- Appendix D includes more details of the experiments.

# A  Proofs for Algorithm 1

We introduce the definition of the (single-step) perturbed transition and reward set, following the definition of $\mathcal{C}(M^\circ, \eta)$.

**Definition 2** ($\eta$-perturbation). *We say that a transition and reward function pair $(T', R')$ at step $h$ is an $\eta$-perturbation of the pair $(T_h, R_h)$ if there exists a function $\xi : \mathcal{S} \times \mathcal{A} \to \Delta(\mathcal{A})$ that for all $s, s'$ and $a$ satisfies $\xi(a \mid s, a) \geq 1 - \eta$ and*

$$T'(s' \mid s, a) = \sum_{a' \in \mathcal{A}} T_h(s' \mid s, a') \xi(a' \mid s, a)$$
$$R'(s, a) = \sum_{a' \in \mathcal{A}} R_h(s, a') \xi(a' \mid s, a).$$

*With slight abusing of the notation, the set of all $\eta$-perturbations of $T_h, R_h$ is denoted $\mathcal{C}((R_h, T_h), \eta)$.*

It is straightforward to see that $(R', T') \in \mathcal{C}((R_h, T_h), \eta)$ if and only if there exists an MDP $M' \in \mathcal{C}(M, \eta)$ such that $(R', T')$ is the $h$-th step reward and transition functions in $M'$ and $(R_h, T_h)$ is the $h$-th step reward and transition functions in $M$.

Now we prove a lemma about the function $\tilde{Q}_h$ used in Algorithm 1.

**Lemma 1.** *For any $h \in [H]$, and $s \in \mathcal{S}_h$, the learned function $\tilde{Q}_h$ in Algorithm 1 satisfies:*

$$(1 - \eta) \max_{a \in \mathcal{A}} \tilde{Q}_h(s, a) + \eta \min_{a \in \mathcal{A}} \tilde{Q}_h(s, a) = \max_{a \in \mathcal{A}} \min_{(R,T) \in \mathcal{C}((R_h^\circ, T_h^\circ), \eta)} \mathrm{E}_{s' \sim T(\cdot|s,a)}[R(s, a) + \tilde{V}_{h+1}(s')]$$

*Proof.* By the definition of $\mathcal{C}((R_h^\circ, T_h^\circ), \eta)$, we know that

$$\min_{(R,T) \in \mathcal{C}((R_h^\circ, T_h^\circ), \eta)} \mathrm{E}_{s' \sim T(\cdot|s,a)}[R(s, a) + \tilde{V}_{h+1}(s')] \tag{2}$$

$$= \min_{(R,T) \in \mathcal{C}((R_h^\circ, T_h^\circ), \eta)} R(s, a) + \sum_{s' \in \mathcal{S}} T(s'|s, a) \tilde{V}_{h+1}(s') \tag{3}$$

$$= \min_{\xi(\cdot|s,a)} \sum_{a' \in \mathcal{A}} \left( \xi(a'|s, a) R_h^\circ(s, a') + \sum_{s' \in \mathcal{S}} \xi(a'|s, a) T_h^\circ(s'|s, a) \tilde{V}_{h+1}(s') \right) \tag{4}$$

$$= \min_{\xi(\cdot|s,a)} \sum_{a' \in \mathcal{A}} \xi(a'|s, a) \left( R_h^\circ(s, a') + \sum_{s' \in \mathcal{S}} T_h^\circ(s'|s, a) \tilde{V}_{h+1}(s') \right) \tag{5}$$

$$= \min_{\xi(\cdot|s,a)} \sum_{a'} \xi(a'|s, a) \tilde{Q}_h(s, a') \tag{6}$$

$$= (1 - \eta) \tilde{Q}_h(s, a) + \eta \min_{a' \in \mathcal{A}} \tilde{Q}_h(s, a'), \tag{7}$$

where $\min$ over $\xi(\cdot|s, a)$ is taken so that it still satisfies the requirement of $\xi(a \mid s, a) \geq 1 - \eta$. This is then used in the last equality, where the minimum is achieved by placing the maximum allowed mass, which is $\eta$, on the action with smallest value of $\tilde{Q}_h(s, a')$ and remaining on $\tilde{Q}_h(s, a)$. Thus we proved that

$$(1 - \eta) \tilde{Q}_h(s, a) + \eta \min_{a'} \tilde{Q}_h(s, a') = \min_{(R,T) \in \mathcal{C}((R_h^\circ, T_h^\circ), \eta)} \mathrm{E}_{s' \sim T(\cdot|s,a)}[R(s, a) + \tilde{V}_{h+1}(s')]$$

The proof is completed by taking maximum over actions on both side. $\square$

**Completing the Proof of Theorem 1**

Now we prove the Theorem 1 as the main result for Algorithm 1.

*Proof.* By Theorem 2.2 in Iyengar [2005], since $Q_h$ and $V_h$ satisfy Lemma 1, we have that $\tilde{V}_1(s) = \sup_{\psi \in \Psi} \min_{M \in \mathcal{C}(M^\circ, \eta)} V_{M,1}^\psi(s)$. The perturbation we defined consists of independent perturbation at different time step $h$, therefore it satisfy the "rectangularity" condition defined in Iyengar [2005].[2]

---

[2]Rectangularity condition requires the choice of perturbation in one time step cannot limit the choice of perturbation in other time steps. We refer to Iyengar [2005] for the formal definition.

From Iyengar [2005], the robust policy is given by

$$\rho_h(s) = \underset{a}{\arg\max} \min_{(R,T) \in \mathcal{C}((R_h^\circ, T_h^\circ), \eta)} \mathbb{E}_{s' \sim T(\cdot|s,a)}[R(s,a) + \tilde{V}_{h+1}(s')] \tag{8}$$

$$= \underset{a}{\arg\max} \left( (1-\eta)\tilde{Q}_h(s,a) + \eta \min_{a' \in \mathcal{A}} \tilde{Q}_h(s,a') \right) \tag{9}$$

$$= \underset{a \in \mathcal{A}}{\arg\max} \tilde{Q}_h(s,a) \tag{10}$$

This is how Algorithm 1 computes the policy it returns. Therefore, Algorithm 1 returns the robust policy $\rho$. $\qquad\square$

## A.1 An alternative Proof for Algorithm 1 from the First Principles

The previous proof relies on a more general results from Iyengar [2005] in a more complicated form. However the proof can be simplified since our perturbation set takes a particular linear structure. So for completeness and readability, we also include another complete proof derived from the first principle. We acknowledge that the proof is heavily inspired by previous work in robust dynamics programming [Bagnell et al., 2001, Iyengar, 2005].

Let $\Psi$ be the set of all mappings from $[H] \times \mathcal{S}$ to $\Delta(\mathcal{A})$. The following proposition follows directly from application of max-min inequality.

**Proposition 1.** $\forall h \in [H], s \in \mathcal{S}$

$$\max_{\psi \in \Psi} \min_{M \in \mathcal{C}(M^\circ, \eta)} V_{M,h}^\psi(s) \le \min_{M \in \mathcal{C}(M^\circ, \eta)} \max_{\psi \in \Psi} V_{M,h}^\pi(s) \coloneqq \min_{M \in \mathcal{C}(M^\circ, \eta)} V_{M,h}^\star(s)$$

Given this, we are going to explain the high level structure of the proof. For $\tilde{V}_h$, we are going to show that

$$\tilde{V}_h(s) \ge \min_{M \in \mathcal{C}(M^\circ, \eta)} V_{M,h}^\star(s) \tag{11}$$

$$\tilde{V}_h(s) \le \min_{M \in \mathcal{C}(M^\circ, \eta)} V_{M,h}^{\tilde{\psi}}(s), \tag{12}$$

for some $\tilde{\psi}$.

Then from Proposition 1, we will have that

$$\tilde{V}_h(s) \le \min_{M \in \mathcal{C}(M^\circ, \eta)} V_{M,h}^{\tilde{\psi}}(s) \le \max_{\psi \in \Psi} \min_{M \in \mathcal{C}(M^\circ, \eta)} V_{M,h}^\psi(s) \le \min_{M \in \mathcal{C}(M^\circ, \eta)} V_{M,h}^\star(s) \le \tilde{V}_h(s) \tag{13}$$

Then we will have that all the inequalities must be equality and the $\tilde{\psi}$ must be the robust policy $\rho$.

Now we will show that Equation 11 and Equation 12 holds for the $\tilde{\psi}$ which is the output from Algorithm 1, in the following two lemmas.

**Lemma 2.** *For any $s \in \mathcal{S}$ and $h \in [H]$, $\tilde{V}_h(s) \ge \min_{M \in \mathcal{C}(M^\circ, \eta)} V_{M,h}^\star(s)$*

*Proof.* For any $h \in [H]$, we will construct a $M \in \mathcal{C}(M^\circ, \eta)$ such that $\forall s, \tilde{V}_h(s) = V_{M,h}^\star(s)$. Then for any $s \in \mathcal{S}$, $\tilde{V}_h(s) = V_{M,h}^\star(s) \ge \min_{M \in \mathcal{C}(M^\circ, \eta)} V_{M,h}^\star(s)$ and we prove this lemma. Now we prove the following statement by induction on $h$. For any $h \in [H+1], \exists M \in \mathcal{C}(M^\circ, \eta)$ such that $\forall s \in \mathcal{S}, \tilde{V}_h(s) = V_{M,h}^\star(s)$.

For the basement of induction, $\tilde{V}_{H+1}(s) = V_{M,h+1}^\star(s) = 0$ for any $M$. Thus the induction statement holds for $h = H + 1$.

Second, let us assume that the induction statement holds for $h + 1$, where $h \in [H]$: $\forall s, \tilde{V}_{h+1}(s) = V_{M_{h+1},h+1}^\star(s)$ for some $M_{h+1} \in \mathcal{C}(M^\circ, \eta)$. Notice that this statement is for the value function at $h + 1$ step and holds for $M_{h+1}$, thus for any $M \in \mathcal{C}(M^\circ, \eta)$ that shares the reward and transition functions on and after step $h+1$, the statement also holds. This is because in our definition of episodic MDP the perturbation class the reward and transition functions at different steps are independent.

Now we are going to construct $M_h$ such that $\tilde{V}_h(s) = V^\star_{M_h,h}(s)$. More specifically, we will only construct the reward and transition functions at $h$-th step: $R_h$ and $T_h$, and concatenate it with other component in $M_{h+1}$.

For any $s$, let $\underline{a}$ be $\operatorname{argmin}_{a \in \mathcal{A}} \tilde{Q}_h(s, a)$. Given the $s$ and $\underline{a}$, construct $T_h(s'|s, a)$ and $R_h(s, a)$ for any $s, a, s'$ as the following.

$$T_h(s'|s, a) = (1 - \eta)T^\circ_h(s'|s, a) + \eta T^\circ_h(s'|s, \underline{a}) \tag{14}$$

$$R_h(s, a) = (1 - \eta)R^\circ_h(s, a) + \eta R^\circ_h(s, \underline{a}) \tag{15}$$

Then we finish the proof of statement for $h$ by

$$\tilde{V}_h(s) = (1 - \eta) \max_a \tilde{Q}_h(s, a) + \eta \min_a \tilde{Q}_h(s, a) \tag{16}$$

$$= (1 - \eta) \max_a \left( R^\circ_h(s, a) + \sum_{s'} T^\circ_h(s'|s, a)\tilde{V}_{h+1}(s') \right) \tag{17}$$

$$+ \eta \left( R^\circ_h(s, \underline{a}) + \sum_{s'} T^\circ_h(s'|s, \underline{a})\tilde{V}_{h+1}(s') \right)$$

$$= \max_a \left( R_h(s, a) + (1 - \eta) \sum_{s'} T^\circ_h(s'|s, a)\tilde{V}_{h+1}(s') + \eta \sum_{s'} T^\circ_h(s'|s, \underline{a})\tilde{V}_{h+1}(s') \right) \tag{18}$$

$$= \max_a \left( R_h(s, a) + \sum_{s'} \left( (1 - \eta)T^\circ_h(s'|s, a) + \eta T^\circ_h(s'|s, \underline{a}) \right) \tilde{V}_{h+1}(s') \right) \tag{19}$$

$$= \max_a \left( R_h(s, a) + \sum_{s'} T_h(s'|s, a)\tilde{V}_{h+1}(s') \right) \tag{20}$$

$$= \max_a \left( R_h(s, a) + \sum_{s'} T_h(s'|s, a)V^\star_{M_{h+1},h+1}(s) \right) \tag{21}$$

$$= V^\star_{M_h,h}(s) \tag{22}$$

By induction, we proved that there exist a $M \in \mathcal{C}(M^\circ, \eta)$ such that $\tilde{V}_h(s) = V^\star_{M,h}(s)$ for any $s \in \mathcal{S}, h \in [H + 1]$. Thus we finish the proof of $\tilde{V}_h(s) \geq \min_{M \in \mathcal{C}(M^\circ, \eta)} V^\star_{M,h}(s)$. $\qquad\square$

**Lemma 3.** *For the policy $\tilde{\psi}$ computed from Algorithm 1, $\forall h \in [H], s \in \mathcal{S}, \min_{M \in \mathcal{C}(M^\circ, \eta)} V^{\tilde{\psi}}_{M,h}(s) \geq \tilde{V}_h(s)$*

*Proof.* We prove it by induction on $h$ from $H + 1$ to 1. In the base case ($h = H + 1$) we have, $\tilde{V}_{H+1}(s) = 0 = V^{\tilde{\psi}}_{M,H+1}(s)$ for all $s \in \mathcal{S}$ and any $M$. This implies: $\forall s, \tilde{V}_{H+1}(s) \leq \min_{M \in \mathcal{C}(M^\circ, \eta)} V^{\tilde{\psi}}_{M,H+1}(s)$.

We now consider the general case. We assume that $\forall s \in \mathcal{S}, \tilde{V}_{h+1}(s) \leq \min_{M \in \mathcal{C}(M^\circ, \eta)} V^{\tilde{\psi}}_{M,h+1}(s)$. Fix any $s \in \mathcal{S}$ and any $M \in \mathcal{C}(M^\circ, \eta)$, we are going to show that $\tilde{V}_h(s) \leq V^{\tilde{\psi}}_{M,h}(s)$. Let $R_h$ and $T_h$ be its reward and transition function at $h$-th step, and $\xi_h$ be the perturbation variables. We have $1 - \xi_h(a \mid s, a) \leq \eta$ by the definition of $M \in \mathcal{C}(M^\circ, \eta)$. So $1 - \xi_h(\tilde{\psi}(s) \mid s, \tilde{\psi}(s)) \leq \eta$.

$$\tilde{V}_h(s) = (1 - \eta) \max_a \tilde{Q}_h(s, a) + \eta \min_a \tilde{Q}_h(s, a) \tag{23}$$

$$= (1 - \eta) \left( R^\circ_h(s, \tilde{\psi}(s)) + \sum_{s'} T^\circ_h(s'|s, \tilde{\psi}(s))\tilde{V}_{h+1}(s') \right) \tag{24}$$

$$+ \eta \left( R^\circ_h(s, \underline{a}) + \sum_{s'} T^\circ_h(s'|s, \underline{a})\tilde{V}_{h+1}(s') \right) \tag{25}$$

$$\leq \xi_h(\tilde{\psi}(s) \mid s, \tilde{\psi}(s)) \left( R_h^\circ(s, \tilde{\psi}(s)) + \sum_{s'} T_h^\circ(s'|s, \tilde{\psi}(s)) \tilde{V}_{h+1}(s') \right)$$

$$+ \left( \sum_{a \neq \tilde{\psi}(s)} \xi_h(a \mid s, \tilde{\psi}(s)) \right) \left( R_h^\circ(s, \underline{a}) + \sum_{s'} T_h^\circ(s'|s, \underline{a}) \tilde{V}_{h+1}(s') \right) \quad (26)$$

$$\leq \xi_h(\tilde{\psi}(s) \mid s, \tilde{\psi}(s)) \left( R_h^\circ(s, \tilde{\psi}(s)) + \sum_{s'} T_h^\circ(s'|s, \tilde{\psi}(s)) \tilde{V}_{h+1}(s') \right)$$

$$+ \sum_{a \neq \tilde{\psi}(s)} \xi_h(a \mid s, \tilde{\psi}(s)) \left( R_h^\circ(s, a) + \sum_{s'} T_h^\circ(s'|s, a) \tilde{V}_{h+1}(s') \right) \quad (27)$$

$$= \sum_a \xi_h(a \mid s, \tilde{\psi}(s)) \left( R_h^\circ(s, a) + \sum_{s'} T_h^\circ(s'|s, a) \tilde{V}_{h+1}(s') \right) \quad (28)$$

$$= R_h(s, \tilde{\psi}(s)) + \sum_{s'} T_h(s'|s, \tilde{\psi}(s)) \tilde{V}_{h+1}(s') \quad (29)$$

$$\leq R_h(s, \tilde{\psi}(s)) + \sum_{s'} T_h(s'|s, \tilde{\psi}(s)) \min_{M' \in \mathcal{C}(M^\circ, \eta)} V_{M',h+1}^{\tilde{\psi}}(s') \quad (30)$$

$$\leq R_h(s, \tilde{\psi}(s)) + \sum_{s'} T_h(s'|s, \tilde{\psi}(s)) V_{M,h+1}^{\tilde{\psi}}(s') = V_{M,h}^{\tilde{\psi}}(s) \quad (31)$$

Since this is for any $M \in \mathcal{C}(M^\circ, \eta)$, we proved that $\tilde{V}_h(s) \leq \min_{M \in \mathcal{C}(M^\circ, \eta)} V_{M,h}^\pi(s)$. □

Now we prove the Theorem 1 as the main result for Algorithm 1.

*Proof.* We have proved Equation 11 and Equation 12 holds for the $\tilde{\psi}$ which is the output from Algorithm 1, in previous two lemmas. Now combine this with the inequality of min max values, we have that

$$\tilde{V}_h(s) \leq \min_{M \in \mathcal{C}(M^\circ, \eta)} V_{M,h}^{\tilde{\psi}}(s) \leq \max_{\psi \in \Psi} \min_{M \in \mathcal{C}(M^\circ, \eta)} V_{M,h}^{\psi}(s) \leq \min_{M \in \mathcal{C}(M^\circ, \eta)} V_{M,h}^{\star}(s) \leq \tilde{V}_h(s) \quad (32)$$

So we will have that all the inequalities must be equality and the $\tilde{\psi}$ output by Algorithm 1 must be the robust policy $\rho$ defined in Equation 1. □

## B   Proofs for Algorithm 2

In this section, we are going to prove a less restrictive and more general version of Theorem 2 stated below. We allow the $\epsilon_\mathcal{F}$ to take any values and the initial state in the true environment has a small probability $\epsilon_0$ when it is not equals to the $s_{\text{init}}$. We show the how the error bounds scales with these approximation errors.

**Theorem 3.** *Let $M^\circ$ be a deterministic abstract simulator. Let $\mathbf{M}^\star$ be a target environment for which $\mathbf{M}^\star$ is an $\eta$-perturbation of $M^\circ$ for some $\eta < 0.5$. Let $\epsilon_0$ be the probability that $s_0 \neq s_{\text{init}}$ in $M$. Let $\mathcal{F}$ be a class of functions realizable with respect to $\mathbf{M}^\star$. Then for any $\epsilon > 0$ and $\delta \in (0, 1)$, Algorithm 2 with oracle access to $\mathbf{M}^\star$, optimization-oracle access to $\mathcal{F}$, and inputs $M^\circ$, $\eta$, $\epsilon$, and $\delta$, executes $n = \mathcal{O}\left( \frac{H^4 |\mathcal{A}|^3 \ln(|\mathcal{F}|/\delta)}{\epsilon (1-2\eta)^2} \right)$ [3] episodes and returns a practicable policy $\pi$ that with probability at least $1 - \delta$ satisfies*

$$V_{\mathbf{M}^\star}^\pi \geq V_{\mathbf{M}^\star}^{\rho \circ \phi^\star} - \epsilon - \frac{4H^4 |\mathcal{A}|^3 \epsilon_\mathcal{F}}{(1-2\eta)^2} - H\epsilon_0.$$

The proof to Theorem 3 has three steps. First we show the accuracy of learned ERM classifier of the classification problem in line 10. Second we show the accuracy of the learned action decoder

---

[3]Notice that here the dependency on horizon is $H^4$ instead of $H^3$ in the main paper. We made a mistake in the statement of this theorem in the main paper.

$\alpha_{h+1}(x_h, x_{h+1})$ under the roll-in distribution of $x_h$. Last the error bound in the theorem can be proved by bounding the union probability of failing to predict the action in each steps. Now we give some lemmas and proofs in these three steps in the following three subsections, followed by the proof of the final sample complexity results.

To prove Theorem 2 in the main paper, we set the $\epsilon$ in Algorithm 2 to be $\epsilon/2$, let $\epsilon_{\mathcal{F}}$ satisfy the assumptions in Theorem 2, and let $\epsilon_0 = 0$. Thus the right hand side in Theorem 3 can be becomes $\epsilon$ and we proved Theorem 2.

### B.1 Accuracy of action classification

First we define some notation that will be used in the proofs for TASID (Algorithm 2). Let the uniform distribution over action space be denoted $\mathtt{Unf}(\mathcal{A})$. We will use $\mathtt{Unf}(a) = 1/|\mathcal{A}|$ to denote the probability over action $a$ given by uniform distribution. We define $P_h(x)$ to be the distribution from which $x_h$ is sampled from in Line 9 in the $h^{th}$ iteration of Algorithm 2. Formally, $P_h(x_h)$ is the probability of receiving observation $x_h$ at time step $h$ when sampling the first $h-1$ actions according to the learned practicable policy $\pi_{1:h-1}$ in the target environment (when $h = 1$, then $x_1$ is directly emitted from the start state, i.e., $x_1 \sim q(\cdot \mid s_{\text{init}})$). Finally, we denote the joint distribution $P_h(x, a, x') := P_h(x) \times \mathtt{Unf}(\mathcal{A}) \times T_h^\star(x'|x, a)$ from which $(x_h, a_h, x_{h+1})$ are sampled from in Line 9.

Now we show the form of conditional distribution of action given two observations, $f_h^\star(a \mid x, x')$, under the joint distribution $P_h(x, a, x')$. Moreover, we prove the form of $f_h^\star(a \mid x, x')$ for any roll-in distribution of the first observation $x$ is the same.

**Lemma 4.** *Given any joint distribution $P(x, a, x') = P(x)\mathtt{Unf}(a)T_h^\star(x' \mid x, a)$, where $P(x)$ is any prior distribution over $x$ and $\mathtt{Unf}(a)$ denotes the uniform distribution over actions, the posterior distribution over action $a$ given pair of observation $x, x'$ is:*

$$f_h^\star(a \mid x, x') = \frac{T_h^\star(x'|x, a)}{\sum_{a'} T_h^\star(x'|x, a')} = \frac{T_h^\star(\phi^\star(x')|\phi^\star(x), a)}{\sum_{a'} T_h^\star(\phi^\star(x')|\phi^\star(x), a')} \tag{33}$$

*Proof.* We start by applying the Bayes' rule.

$$f_h^\star(a \mid x, x') = \frac{P(x, a, x')}{\sum_{a' \in \mathcal{A}} P(x, a', x')} = \frac{P(x)\mathtt{Unf}(a)T_h^\star(x'|x, a)}{\sum_{a' \in \mathcal{A}} P(x)\mathtt{Unf}(a')T_h^\star(x'|x, a')} \tag{34}$$

$$= \frac{P(x)/|\mathcal{A}|T_h^\star(x'|x, a)}{P(x)/|\mathcal{A}| \sum_{a' \in \mathcal{A}} T_h^\star(x'|x, a')} \tag{35}$$

$$= \frac{T_h^\star(x'|x, a)}{\sum_{a' \in \mathcal{A}} T_h^\star(x'|x, a')} \tag{36}$$

$$= \frac{T_h^\star(\phi^\star(x') \mid \phi^\star(x), a)}{\sum_{a' \in \mathcal{A}} T_h^\star(\phi^\star(x') \mid \phi^\star(x), a')}, \tag{37}$$

where the last equality uses $T_h^\star(x' \mid x, a') = q(x' \mid \phi^\star(x'))T_h^\star(\phi^\star(x') \mid \phi^\star(x), a')$ for all $a' \in \mathcal{A}$, which follows from the block MDP structure of $M^\star$. $\square$

Note that, in particular, Lemma 4 holds for the joint distribution $P_h(x, a, x')$ that we described earlier.

Now we can show the error bound of empirical log-likelihood maximizer. Before that, we introduce some useful lemmas from Agarwal et al. [2020] for the completeness of this paper.

**Lemma 5** (Lemma 24 in Agarwal et al. [2020]). *Let $\mathcal{D}$ be a dataset of $n$ $(x, a, x')$ samples, and let $\mathcal{D}'$ be another independent dataset from the same distribution. Let $L(f, \mathcal{D}) = \sum_{i=1}^n \ell(f, (x_i, y_i))$ be any function that decomposes additively across examples where $\ell$ is any function, and let $\hat{f}(\mathcal{D})$ be any estimator taking as input random variable $\mathcal{D}$ and with range $\mathcal{F}$. Then, with probability 1- $\delta$,*

$$-\log \mathrm{E}_{\mathcal{D}'} \exp(L(\hat{f}(\mathcal{D}), \mathcal{D}')) \leq -L(\hat{f}(\mathcal{D}), \mathcal{D}) + \log|\mathcal{F}| + \log(1/\delta)$$

**Lemma 6** (Lemma 25 in Agarwal et al. [2020]). *For any two conditional probability densities $f_1$, $f_2$ and any distribution $D \in \Delta(\mathcal{X} \times \mathcal{X})$ we have*

$$\mathrm{E}_{x, x' \sim D} \|f_1(\cdot|x, x') - f_2(\cdot|x, x')\|_{TV}^2 \leq -2 \log \mathrm{E}_{x, x' \sim D, a \sim f_2(\cdot|x, x')} \exp\left(-\frac{1}{2} \log \frac{f_2(a|x, x')}{f_1(a|x, x')}\right)$$

The proof of the empirical log-likelihood maximizer's error bound follows the proof in Agarwal et al. [2020] as well. However we modified the proof to only assume approximate realizability assumption instead of perfect realizaibility.

**Theorem 4.** *Let $\mathcal{D}_h$ be a data set with $n_D$ transitions (x,a,x') sampled i.i.d. from $P_h(x, a, x')$. Let $f_h(a \mid x, x')$ and $\widetilde{f}_h(a \mid x, x')$ denote the maximizers of empirical log-likelihood and expected log-likelihood respectively:*

$$\widetilde{f}_h(a \mid x, x') := \underset{f \in \mathcal{F}}{\operatorname{argmax}} \operatorname{E}_{x,a,x' \sim P_h}[\ln f(a|x, x')] \tag{38}$$

$$f_h(a \mid x, x') := \underset{f \in \mathcal{F}}{\operatorname{argmax}} \sum_{(x_h, a_h, x_{h+1}) \in \mathcal{D}_h} \ln f(a_h \mid x_h, x_{h+1}) \tag{39}$$

*Then for any $\delta \in (0, 1)$, with probability at least $1 - \delta$ we have that:*

$$\operatorname{E}_{x,x' \sim P_h} \left[ \| f_h(\cdot | x, x') - f_h^\star(\cdot | x, x') \|_{TV}^2 \right] \leq \frac{2 \log(|\mathcal{F}|/\delta)}{n_D} + \epsilon_{\mathcal{F}} \tag{40}$$

*Proof.* We set $L(f, \mathcal{D})$ in Lemma 5 to be $\sum_{(x,a,x') \in \mathcal{D}} -\frac{1}{2} \log \frac{\widetilde{f}_h(a|x,x')}{f_h(a|x,x')}$ where $\mathcal{D}$ is $\mathcal{D}_h$ and $\mathcal{D}'$ to be another I.I.D. dataset with $n_D$ samples drawn independently with $\mathcal{D}$. With this choice, the right hand side is

$$\sum_{(x,a,x') \in \mathcal{D}} \frac{1}{2} \log \frac{\widetilde{f}_h(a|x,x')}{f_h(a|x,x')} + \log |\mathcal{F}| + \log(1/\delta) \leq \log |\mathcal{F}| + \log(1/\delta) \tag{41}$$

since $f_h$ is the empirical maximum likelihood estimator and $f_h^\star \in \mathcal{F}$. The left hand side is

$$- \log \operatorname{E}_{\mathcal{D}'} \left[ \exp \left( \sum_{(x,a,x') \in \mathcal{D}'} -\frac{1}{2} \log \frac{\widetilde{f}_h(a|x,x')}{f_h(a|x,x')} \right) \mid \mathcal{D} \right] \tag{42}$$

$$= - n_D \log \operatorname{E}_{x,a,x' \sim P_h} \exp \left( -\frac{1}{2} \log \frac{\widetilde{f}_h(a|x,x')}{f_h(a|x,x')} \right) \tag{43}$$

$$= - n_D \log \operatorname{E}_{P_h} \exp \left( -\frac{1}{2} \log \frac{f_h^\star(a|x,x')}{f_h(a|x,x')} - \frac{1}{2} \log \frac{\widetilde{f}_h(a|x,x')}{f_h^\star(a|x,x')} \right) \tag{44}$$

$$= - n_D \log \operatorname{E}_{P_h} \left[ \exp \left( -\frac{1}{2} \log \frac{f_h^\star(a|x,x')}{f_h(a|x,x')} \right) \cdot \sqrt{\frac{f_h^\star(a|x,x')}{\widetilde{f}_h(a|x,x')}} \right] \tag{45}$$

By Assumption 3, we have that $\sqrt{\frac{f_h^\star(a|x,x')}{\widetilde{f}_h(a|x,x')}}$ is upper bounded by $\sqrt{1 + \epsilon_{\mathcal{F}}}$ uniformly overall $x, a, x'$. So we have that

$$- \log \operatorname{E}_{\mathcal{D}'} \left[ \exp \left( \sum_{(x,a,x') \in \mathcal{D}'} -\frac{1}{2} \log \frac{\widetilde{f}_h(a|x,x')}{f_h(a|x,x')} \right) \mid \mathcal{D} \right] \tag{46}$$

$$\geq - n_D \log \operatorname{E}_{P_h} \left[ \exp \left( -\frac{1}{2} \log \frac{f_h^\star(a|x,x')}{f_h(a|x,x')} \right) \right] - n_D \log \sqrt{1 + \epsilon_{\mathcal{F}}} \tag{47}$$

$$\geq \frac{n_D}{2} \operatorname{E}_{x,x' \sim P_h} \left[ \| f_h(\cdot | x, x') - f_h^\star(\cdot | x, x') \|_{TV}^2 \right] - n_D \log \sqrt{1 + \epsilon_{\mathcal{F}}} \tag{48}$$

The last steps follows from Lemma 6. After combine this with the right hand side and rearrange the terms, we have

$$\operatorname{E}_{x,x' \sim P_h} \left[ \| f_h(\cdot | x, x') - f_h^\star(\cdot | x, x') \|_{TV}^2 \right] \leq \frac{2 \log(|\mathcal{F}|/\delta)}{n_D} + \log(1 + \epsilon_{\mathcal{F}}) \tag{49}$$

$$\leq \frac{2 \log(|\mathcal{F}|/\delta)}{n_D} + \epsilon_{\mathcal{F}} \tag{50}$$

$\square$

## B.2 One-step accuracy of action decoder

In the last subsection, we have showed the learned function $f_h$ approaches the posterior distribution of action at a rate of $1/n_D$. In Algorithm 2, the learned function $f_h$ is used to build the state decoder $\alpha_h$. This state decoding process relies on identified the correct "shadow" actions.

In this section we first show that there is a separation between the correct shadow action's probability and random actions' probabilities, in the posterior distribution of action. This relies on the transition dynamics in target environment lies in $\mathcal{C}(T^\circ, \eta)$. Then we show the action decoding accuracy. The idea to bound the 1-step action decoding accuracy is to apply the Markov's inequality on the event that the $\arg\max$ action of $f_h$ is wrong.

First, we define the set of shadow actions given two successive states.

**Definition 3** (Shadow action set). *For any states pair $(s, s')$, we define a set of actions $A_h(s, s') = \{a \in \mathcal{A} : T_h^\circ(s'|s, a) = 1\}$ and $A_h^c(s, s') = \mathcal{A} \backslash A_h(s, s')$.*

*Since the observation emission function $q^\star$ maps different $s$ to disjoint observation subspace, the shadow action sets can also be defined on the corresponding observation pairs $x, x'$. We define $A_h(x, x') = \{a \in \mathcal{A} : T_h^\circ(\phi^\star(x')|\phi^\star(x), a) = 1\}$, and similarly for $A_h^c(x, x')$*

By definition, we have that $A_h(x, x') = A_h(s, s')$ if $x \sim q^\star(\cdot|s)$ and $x' \sim q^\star(\cdot|s')$. So later we may use these two notations interchangeably for convenience.

Next we prove the accuracy result of decoder $\alpha$ under the joint distribution with *any* practicable policy, but learned from the dataset from *uniform* action distribution. We use $\mathbb{1}$ as an indicator function of random events.

**Lemma 7** (Accuracy of decoder). *For any practicable policy $\pi$, let $P_{h,\pi}(x, a, x') := P_h(x)\pi(a|x)T^\star(x'|x, a)$ be the joint distribution of $P_h(x)$, policy $\pi$ and transition function $T^\star$.*

*Let $\mathcal{D}_h$ be a data set with $n_D$ transitions (x,a,x') sampled i.i.d. from $P_h(x, a, x')$. For any $h \in [H]$ and $\delta \in (0, 1)$, and any practicable policy $\pi$, with probability at least $1 - \delta$, we have that*

$$\mathrm{E}_{P_{h,\pi}}\left[\mathbb{1}(\alpha_h(x, x') \in A_h(x, x'))\right] \geq 1 - \frac{8h|\mathcal{A}|^3 \ln(|\mathcal{F}|/\delta)}{n_D(1 - 2\eta)^2} \tag{51}$$

*Proof.* For any $a_1 \in A_h(x, x')$, and $a_2 \in A_h^c(x, x')$,

$$f_h^\star(a_1 \mid x, x') - f_h^\star(a_2 \mid x, x') = \frac{T_h^\star(\phi^\star(x') \mid \phi^\star(x), a_1) - T_h^\star(\phi^\star(x') \mid \phi^\star(x), a_2)}{\sum_{a' \in \mathcal{A}} T_h^\star(\phi^\star(x') \mid \phi^\star(x), a')} \tag{52}$$

$$\geq \frac{(1 - \eta) - \eta}{\sum_{a' \in \mathcal{A}} T_h^\star(\phi^\star(x') \mid \phi^\star(x), a')} \geq \frac{(1 - 2\eta)}{|\mathcal{A}|} \tag{53}$$

Given that the gap is $(1-2\eta)/|\mathcal{A}|$, we have that for any fixed $x, x'$, if $\|f_h(\cdot, x, x') - f_h^\star(\cdot, x, x')\|_{\mathrm{TV}} < (1-2\eta)/2|\mathcal{A}|$, then $\arg\max f_h(\cdot, x, x') \in A_h(x, x')$.

Recall that $\alpha_h(x, x') = \arg\max_{a \in \mathcal{A}} f_h(a \mid x, x')$. We have

$$\mathrm{E}_{P_h}[\mathbb{1}(\alpha_h(x, x') \notin A_h(x, x'))] = \Pr(\alpha_h(x, x') \notin A_h(x, x')) \tag{54}$$

$$\leq \Pr\left(\|f_h^\star(\cdot \mid x, x') - f_h(\cdot \mid x, x')\|_{\mathrm{TV}} \geq \frac{(1 - 2\eta)}{2|\mathcal{A}|}\right) \tag{55}$$

$$= \Pr\left(\|f_h^\star(\cdot \mid x, x') - f_h(\cdot \mid x, x')\|_{\mathrm{TV}}^2 \geq \frac{(1 - 2\eta)^2}{4|\mathcal{A}|^2}\right) \tag{56}$$

$$\leq \frac{\mathrm{E}\left[\|f_h^\star(\cdot \mid x, x') - f_h(\cdot, |x, x')\|_{\mathrm{TV}}^2\right]}{(1-2\eta)^2/4|\mathcal{A}|^2} \tag{57}$$

$$\leq \frac{8|\mathcal{A}|^2 \ln(|\mathcal{F}|/\delta)}{n_D(1 - 2\eta)^2} + \frac{4|\mathcal{A}|^2 \epsilon_{\mathcal{F}}}{(1 - 2\eta)^2} \tag{58}$$

The second to last step follows from Markov's inequality, and the last step follows from Theorem 4.

Notice that this is the error under distribution of uniform action $P_h(x, a, x') := P(x) \circ \mathtt{Unf} \circ T_h^\star$. For any practicable policy $\pi$, since $\frac{P_{h,\pi}(x,a,x')}{P_h(x,a,x')} = \frac{\pi(a|x)}{1/|\mathcal{A}|} \leq |\mathcal{A}|$,

$$\mathrm{E}_{P_{h,\pi}}\left[\mathbb{1}(\alpha_h(x, x') \notin A_h(x, x'))\right] \leq \frac{8|\mathcal{A}|^3 \ln(|\mathcal{F}|/\delta)}{n_D(1 - 2\eta)^2} + \frac{4|\mathcal{A}|^3 \epsilon_\mathcal{F}}{(1 - 2\eta)^2} \tag{59}$$

Then taking the complement of the event in the indicator function finished the proof. $\square$

### B.3  Analysis of the sample complexity

Now we are going prove that the learned policy recover the input latent policy with high probability.

**Lemma 8.** *For any $h \in [H]$, let $\mathbf{x}_{1:h+1} := x_1, x_2, \ldots, x_{h+1}$ and $\mathbf{a}_{1:h} := a_1, a_2, \ldots, a_h$ be the state and action sequence generated from $(\pi_1, \ldots, \pi_h)$ output by Algorithm 2. Given the high probability event in Theorem 4, we have that for any $h \in [H]$*

$$\mathrm{E}_{\pi_{1:h}}\left[\sum_{k=1}^h \mathbb{1}\{a_k \neq \rho(\phi(x_k))\}\right] \leq \frac{8h^2|\mathcal{A}|^3 \ln(|\mathcal{F}|/\delta)}{n_D(1 - 2\eta)^2} + \frac{4h^2|\mathcal{A}|^3 \epsilon_\mathcal{F}}{(1 - 2\eta)^2} + \epsilon_0 \tag{60}$$

*Proof.* If the action decoder $\alpha_k$ is correct, i.e. $\alpha_k(x_k, x_{k+1}) \in A_k(x_k, x_{k+1})$ for any $k \in [h]$. Then if the initial state is $s_{\mathrm{init}}$, we have that for any $k \in [h]$, $\phi_k(\mathbf{x}_{1:k}) = \phi^\star(x_{k+1})$ due to the determinism of the abstract simulator. Then for any $k \in [h]$,

$$a_k = \pi_k(\mathbf{x}_{1:k}) = \rho(\phi_k(\mathbf{x}_{1:k})) = \rho(\phi^\star(x_k)) \tag{61}$$

That means if we have $a_k \neq \rho(\phi^\star(x_k))$, we must have $\alpha_k(x_k, x_{k+1}) \notin A_k(x_k, x_{k+1})$ for some $j \leq k$, or the initial state is not $s_{\mathrm{init}}$. So under the condition that the initial state is $s_{\mathrm{init}}$,

$$\mathbb{1}\{a_k \neq \rho(\phi^\star(x_k))\} \leq \sum_{j=1}^k \mathbb{1}\{\alpha_j(x_j, x_{j+1}) \notin A_j(x_j, x_{j+1})\} \tag{62}$$

Thus we have

$$\mathrm{E}_{\pi_{1:h}}\left[\sum_{k=1}^h \mathbb{1}\{a_k \neq \rho(\phi^\star(x_k))\}\right] \tag{63}$$

$$= \mathrm{E}_{\pi_{1:h}}\left[\sum_{k=1}^h \sum_{j=1}^k \mathbb{1}\{\alpha_j(x_j, x_{j+1}) \notin A_j(x_j, x_{j+1})\}\right] + \epsilon_0 \tag{64}$$

$$\leq \mathrm{E}_{\pi_{1:h}}\left[h \sum_{j=1}^h \mathbb{1}\{\alpha_j(x_j, x_{j+1}) \notin A_j(x_j, x_{j+1})\}\right] + \epsilon_0 \tag{65}$$

$$\leq \frac{8h^2|\mathcal{A}|^3 \ln(|\mathcal{F}|/\delta)}{n_D(1 - 2\eta)^2} + \frac{4h^2|\mathcal{A}|^3 \epsilon_\mathcal{F}}{(1 - 2\eta)^2} + \epsilon_0 \tag{Lemma 7}$$

$\square$

This immediately gives the following theorem by letting $h = H$ and bounding the value gap for non-optimal actions by $H$. The proof of Theorem 2 follows from this.

*Proof.* We first bound the value gap using preivous lemma and the Performance Difference Lemma [Kakade, 2003].

$$V_{\mathbf{M}^\star}^\pi - v_{\mathbf{M}^\star}^{\rho \circ \phi^\star} \leq \mathrm{E}_{x_h, a_h \sim \pi} \left[ \sum_{h=1}^{H} Q_{\mathbf{M}^\star, h}^{\rho \circ \phi^\star}(x_h, a_h) - V_{\mathbf{M}^\star, h}^{\rho \circ \phi^\star}(x_h) \right] \qquad \text{(Performance Difference)}$$

$$= \mathrm{E}_{s_h, a_h \sim \pi} \left[ \sum_{h=1}^{H} Q_{M^\star, h}^{\rho}(s_h, a_h) - V_{M^\star, h}^{\rho}(s_h) \right] \qquad \text{(Block MDP)}$$

$$\leq \mathrm{E}_{s_h, a_h \sim \pi} \left[ \sum_{h=1}^{H} H \, \mathbb{1}\{a_h \neq \rho(\phi(x_h))\} \right] \qquad \text{(Any } s_h, a_h, Q_h^\rho(s_h, a_h) \in [0, H])$$

$$\leq \frac{8H^3 |\mathcal{A}|^3 \ln(|\mathcal{F}|/\delta)}{n_D (1 - 2\eta)^2} + H\epsilon_0 \qquad \text{(Lemma 8 for } h = H)$$

$$= \frac{8H^4 |\mathcal{A}|^3 \ln(|\mathcal{F}|/\delta)}{n(1 - 2\eta)^2} + \frac{4H^4 |\mathcal{A}|^3 \epsilon_{\mathcal{F}}}{(1 - 2\eta)^2} + H\epsilon_0 \qquad (n = Hn_D)$$

Finally, we can solve the sample complexity by denote the value gap $\epsilon$, and finish the proof. □

## C    Analysis with Stochastic Initial State

We can extend `TASID` to a more general setting where initial state can be stochastically chosen, instead of being deterministic. This setting captures problems such as navigation in a set of house simulators, where dynamics of each house simulator can be deterministic but the choice of initial state, i.e., choice of current house and position of the agent inside the house, can be stochastically chosen.

As Algorithm 1 does not rely on the deterministic initial state, therefore, we only need to show that we can extend Algorithm 2 to the stochastic initial state setting, and find the robust policy learned by Algorithm 1.

First, we introduce the difference in problem settings and our main results under this setting formally. We assume that in both abstract simulator $M^\circ$ and the target environment $\mathbf{M}^\star$, the initial states are sampled from the same distribution $\mu$ over a finite set of initial states $\mathcal{S}_1$ of size $|\mathcal{S}_1| = N$.

We make an assumption that each initial state occurs with a reasonable probability and has a different transition dynamics.

**Assumption 4.** *(Conditions on initial state) For all initial states $s \in \mathcal{S}_1$, we assume $\mu(s) \geq \mu_{min}$ for some $\mu_{min} \in (0, 1]$. Further, there exists a margin $\Gamma > 0$ such that for any two initial states $s, \tilde{s} \in \mathcal{S}_1$ we have:*

$$\|T(\cdot \mid s, a) - T(\cdot \mid \tilde{s}, a)\|_{\mathrm{TV}} \geq \Gamma.$$

Informally, this assumption is required so that we can visit each initial state sufficiently, and use the margin assumption to learn an accurate decoder to cluster initial states. However, note that we cannot directly cluster in the observation space since we do not want to make any additional structural assumptions on it. In contrast, we will use a function-approximation approach where we interact with the observation via a function class.

We present a variation of `TASID` in Algorithm 3 that can address stochastic initial state. The algorithm assumes access to an approximate decoder $\hat{\phi} : \mathcal{X} \to \mathbb{N}$ that can cluster observations from the same initial state together. This decoder can be thought of partitioning the observation space into *decoder states* which recover the true initial states upto relabeling. In Section C.1 we discuss how to learn this decoder using the Homer algorithm Misra et al. [2020]. Algorithm 3 learns a mapping from these decoder states to initial states in $\mathcal{S}_1$ by performing a hypothesis testing algorithm that uses Algorithm 2 in the main paper as a sub-routine. In Section C.2 we discuss this hypothesis test.

We will prove that under a realizability assumption (stated later), this algorithm has the following guarantee.

---

**Algorithm 3** TASID with multiple initial states

---

1: **Input:** An approximate clustering function of initial states $\hat{\phi}$, number of initial state $N$, error bound $\epsilon$, failure probability $\delta$.
2: Initialize counter $\texttt{step}(i) \leftarrow 0$ for $i \in [N]$
3: Initialize state map $\texttt{map}(i) \leftarrow -1$ for $i \in [N]$
4: **for** episode $e = 1, \dots$ **do**
5:      For the initial observation $x$, decode the state by $\hat{\phi}(x)$.
6:      **if** $\texttt{map}(\hat{\phi}(x)) < 0$ **then**
7:          $\texttt{map}(\hat{\phi}(x)) \leftarrow \text{InitialStateTest}(\hat{\phi}(x), \texttt{step}(\hat{\phi}(x)), N, \epsilon, \delta)$
8:          $\texttt{step}(\hat{\phi}(x)) \leftarrow \texttt{step}(\hat{\phi}(x)) + 1$
9:      **else**
10:          Run TASID on $\texttt{map}(\hat{\phi}(x))$

---

---

**Algorithm 4** InitialStateTest

---

1: **Input:** state cluster $i$, global episode counter $t$, number of initial state $N$, error bound $\epsilon$, failure probability $\delta$.
2: **if** t = 0 **then**
3:      Hypothetic state $s \leftarrow 0$
4:      Initialize counter $\texttt{cnt}(i) \leftarrow 0$ for $i \in [N]$
5:      Initialize value estimates $v(i) \leftarrow 0$ for $i \in [N]$
6:      $n_l \leftarrow \frac{8H^4|\mathcal{A}|^3 \ln(N^2|\mathcal{F}|/\delta)}{\epsilon(1-2\eta)^2}$
7:      $n_t \leftarrow \frac{H^2 \ln(N/\delta)}{2\epsilon^2}$
8:      The algorithm instance will maintain the hypothesis state $s$, an episodes counter $\texttt{cnt}(\cdot)$ and a value logger $v(\cdot)$ for the same state cluster $i$ across calls.
9: **if** $\texttt{cnt}(s) < n_l$ **then**
10:      Run TASID for one episode on initial state $s$
11:      $\texttt{cnt}(s) \leftarrow \texttt{cnt}(s) + 1$
12:      **return** -1
13: **else if** $n_l \geq \texttt{cnt}(s) < n_l + n_t$ **then**
14:      Rollout learned policy $\pi$ for one episode and update value average $v(s)$
15:      $\texttt{cnt}(s) \leftarrow \texttt{cnt}(s) + 1$
16:      **return** -1
17: **else**
18:      $s \leftarrow s + 1$
19:      **if** $s = N + 1$ **then**
20:          Run TASID for one episode on initial state $\text{argmax}_s v(s)$
21:          return $\text{argmax}_s v(s)$
22:      **else**
23:          **return** -1

---

**Theorem 5.** *With the assumption about $\epsilon_{\mathcal{F}}$ in Theorem 2, Algorithm 3 will execute a policy that is close to robust policy by $\epsilon$ in all but* $\text{poly} \left\{ N, H, A, \frac{1}{\epsilon}, \frac{1}{(1-2\eta)}, \frac{1}{\mu_{min}} \ln\{\frac{1}{\delta}\} \right\}$ *episodes with probability at least $1 - \delta$.*

Note that when there is a deterministic initial state, i.e., $N = 1$ and $\mu_{\min} = 1$, we recover dependence on the same set of parameters as our main result. For some problems, $N$ maybe significantly smaller than the set of all states. For example, a robot may start an episode from its charging station and there maybe a small number of charging stations in the environment. For these problems, the dependence on $N$ may be acceptable.

In the second part, we propose a hypothesis testing based algorithm, that use Algorithm 2 in the main paper as a sub-routine, and prove the new sample complexity.

### C.1 Learning decoder initial states

We use the Homer algorithm [Misra et al., 2020] to learn the decoder $\hat{\phi}$. We briefly describe the application of this algorithm for time step $h = 1$. Homer collects a dataset $\mathcal{D}$ of $n$ quads as follows: we sample $y$ uniformly in $\{0, 1\}$ and collect two independent transitions $(x^{(1)}, a^{(1)}, x'^{(1)}), (x^{(2)}, a^{(2)}, x'^{(2)})$ at the first time step by taking actions $a^{(1)}$ and $^{(2)}$ uniformly. If $y = 1$ then we add $(x^{(1)}, a^{(1)}, x'^{(1)}, y)$ to $\mathcal{D}$, otherwise, we add $(x^{(1)}, a^{(1)}, x'^{(2)}, y)$. Note that $(x^{(1)}, a^{(1)}, x'^{(2)})$ is an unobserved transition, therefore, we call it an imposter transition, whereas, $(x^{(1)}, a^{(1)}, x'^{(1)})$ is a real transition. We know that there are exactly $N$ initial states since we have access to the simulator. Given a bottleneck function class $\Phi : \{\phi : \mathcal{X} \to [N]\}$ and another regressor class $\mathcal{G} : \{f : [N] \times \mathcal{A} \times \mathcal{X} \to [0, 1]\}$, we train a model to differentiate between real and imposter transition as follows:

$$\hat{g}, \hat{\phi} = \arg \min_{g \in \mathcal{G}, \phi \in \Phi} \sum_{(x, a, x', y) \in \mathcal{D}} \left( g(\phi(x), a, x') - y \right)^2 .$$

**Difference from Misra et al. [2020].** While our approach and analysis in this subsection closely follows Misra et al. [2020], we differ from them in two crucial ways. Firstly, we apply bottleneck on $x$ instead of $x'$, since we want to recover a decoder for initial states. Secondly, Misra et al. [2020] do not assume a margin assumption $\Gamma$ since their approach does not concern with recovering an exact decoder, but only in learning a good set of policies for exploration.

We will denote the function class $\mathcal{G} \circ \Phi = \{g \circ \phi : (x, a, x') \mapsto g(\phi(x), a, x') \mid g \in \mathcal{G}, \phi \in \Phi\}$. Let $D(x, a, x')$ be the marginal distribution over real and imposter transitions, and let $\rho(x') = \mathbb{E}_{x \sim \mu, a \sim \text{unf}(\mathcal{A})} [T(x' \mid x, a)]$ be the marginal probability over $x'$ for real transitions where $\mu$ is the initial state distribution. It can be shown that:

$$D(x, a, x') = \frac{\mu(x)}{2|\mathcal{A}|} \{T(x' \mid x, a) + \rho(x')\}, \tag{66}$$

where $T(x'|x, a)\mu(x)/|\mathcal{A}|$ is the probability of observing a real transition $(x, a, x')$ and $\rho(x')\mu(x)/|\mathcal{A}|$ is the probability of observing an imposter transition $(x, a, x')$ and the factor of $1/2$ comes due to uniform selection over real and imposter transition.

Misra et al. [2020] showed that the Bayes optimal classifier of the prediction problem is given by:

**Lemma 9** (Bayes Optimal Classifier). *For any $(x, a, x')$ in support of $D$, we have:*

$$g^\star(x, a, x') = \frac{T(x' \mid x, a)}{T(x' \mid x, a) + \rho(x')} = \frac{T(\phi^\star(x') \mid \phi^\star(x), a)}{T(\phi^\star(x') \mid \phi^\star(x), a) + \rho(\phi^\star(x'))}$$

*Proof.* See Lemma 9 of Misra et al. [2020]. □

Similar to Misra et al. [2020], we make a realizability assumption stated below that allows us to solve the classification problem well.

**Assumption 5** (Realizability). *We assume that $g^\star \in \mathcal{G} \circ \Phi$.*

We can use the realizability assumption to get the following generalization bound guarantee: for any $\delta \in (0, 1)$ we have:

$$\mathbb{E}_{x, a, x' \sim D} \left[ \left| \hat{g}(\hat{\phi}(x), a, x') - g^\star(x, a, x') \right| \right] \leq \Delta := \sqrt{\frac{C(\mathcal{G} \circ \Phi)}{n} \ln\left(\frac{1}{\delta}\right)}, \tag{67}$$

with probability at least $1 - \delta$, where $C(\mathcal{G} \circ \Phi)$ is a complexity measure for class $\mathcal{G} \circ \Phi$ such as $\ln(|\mathcal{G}||\Phi|)$ or Rademacher complexity. For proof see Proposition 11 and Corollary 6 in Misra et al. [2020]. Note that even though their proof uses a bottleneck model $\phi$ on $x'$ instead of $x$, essentially the same argument holds by symmetry.

**Coupling Distribution.** We define a coupling distribution as

$$D_c(x_1, x_2, a, x') = D(x_1 \mid a, x')D(x_2 \mid a, x')\frac{1}{|\mathcal{A}|}\rho(x'), \tag{68}$$

where $D(x_1 \mid a, x')$ is the conditional distribution derived from the joint distribution $D(x_1, a, x')$ defined earlier. We also define the marginal distribution $D(a, x')$ which gives us $D(x_1 \mid a, x') = D(x_1, a, x') / D(a, x')$.

We present some result related to the distributions defined above that will be useful later for proving important results later.

$$D(a, x') = \sum_x D(x, a, x') = \sum_x \frac{\mu(x)}{2|\mathcal{A}|} \{T(x' \mid x, a) + \rho(x')\} \geq \frac{\rho(x')}{2|\mathcal{A}|}, \tag{69}$$

$$\sum_a D(a, x') = \rho(x'). \tag{70}$$

Using Equation 69 we can prove:

$$\sum_{x_1} D_c(x_1, x_2, a, x') = \sum_{x_1} D(x_1 \mid a, x') \frac{D(x_2, a, x')}{D(a, x')} \frac{\rho(x')}{|\mathcal{A}|} \tag{71}$$

$$\leq 2 \sum_{x_1} D(x_1 \mid a, x') D(x_2, a, x') = 2D(x_2, a, x'). \tag{72}$$

Similarly, we can prove:

$$\sum_{x_2} D_c(x_1, x_2, a, x') \leq 2D(x_1, a, x'). \tag{73}$$

Further, we have:

$$D(x \mid a, x') = \frac{\mu(x)\{T(x' \mid x, a) + \rho(x')\}}{2|\mathcal{A}|D(a, x')} \geq \frac{\mu(x)\rho(x')}{2|\mathcal{A}|D(a, x')} \geq \frac{\mu(x)\rho(x')}{2|\mathcal{A}|\sum_{a \in \mathcal{A}} D(a, x')} = \frac{\mu(x)}{2|\mathcal{A}|},$$

which gives us:

$$\frac{D(x \mid a, x')\rho(x')}{T(x' \mid x, a) + \rho(x')} = \frac{\mu(x)\rho(x')}{2|\mathcal{A}|D(a, x')} \geq \frac{\mu(x)}{2|\mathcal{A}|}. \tag{74}$$

We now state a useful lemma.

**Lemma 10.** *Fix $\delta \in (0, 1)$. Then with probability at least $1 - \delta$ we have*

$$\mathrm{E}_{x_1, x_2, a, x' \sim D_c} \left[ \mathbf{1}\{\hat{\phi}(x_1) = \hat{\phi}(x_2)\} |g^\star(x_1, a, x') - g^\star(x_2, a, x')| \right] \leq 4\Delta.$$

*Proof.* We use triangle inequality to decompose the left hand side as:

$$\mathrm{E}_{(x_1, x_2, a, x') \sim D_c} \left[ \mathbf{1}\{\hat{\phi}(x_1) = \hat{\phi}(x_2)\} |g^\star(x_1, a, x') - g^\star(x_2, a, x')| \right]$$

$$\leq \mathrm{E}_{(x_1, x_2, a, x') \sim D_c} \left[ \mathbf{1}\{\hat{\phi}(x_1) = \hat{\phi}(x_2)\} |g^\star(x_1, a, x') - \hat{g}(\hat{\phi}(x_1), a, x')| \right] +$$

$$\mathrm{E}_{(x_1, x_2, a, x') \sim D_c} \left[ \mathbf{1}\{\hat{\phi}(x_1) = \hat{\phi}(x_2)\} |\hat{g}(\hat{\phi}(x_1), a, x') - g^\star(x_2, a, x')| \right]$$

The first term is bounded as shown below:

$$\mathrm{E}_{(x_1, x_2, a, x') \sim D_c} \left[ \mathbf{1}\{\hat{\phi}(x_1) = \hat{\phi}(x_2)\} |g^\star(x_1, a, x') - \hat{g}(\hat{\phi}(x_1), a, x')| \right]$$

$$\leq \mathrm{E}_{(x_1, x_2, a, x') \sim D_c} \left[ |g^\star(x_1, a, x') - \hat{g}(\hat{\phi}(x_1), a, x')| \right]$$

$$\leq 2\mathrm{E}_{(x, a, x') \sim D} \left[ |g^\star(x, a, x') - \hat{g}(\hat{\phi}(x), a, x')| \right] = 2\Delta,$$

where the second inequality uses Equation 73 and Equation 67. The second term is bounded as:

$$\mathrm{E}_{(x_1, x_2, a, x') \sim D_c} \left[ \mathbf{1}\{\hat{\phi}(x_1) = \hat{\phi}(x_2)\} |\hat{g}(\hat{\phi}(x_1), a, x') - g^\star(x_2, a, x')| \right]$$

$$= \mathrm{E}_{(x_1, x_2, a, x') \sim D_c} \left[ \mathbf{1}\{\hat{\phi}(x_1) = \hat{\phi}(x_2)\} |\hat{g}(\hat{\phi}(x_2), a, x') - g^\star(x_2, a, x')| \right] \leq 2\Delta,$$

where the inequality results from following similar steps used for bounding the first term. Adding the two upper bounds we get $4\Delta$. $\qquad\square$

Using Lemma 9, we have for every $x_1, x_2, x' \in \mathcal{X}, a \in \mathcal{A}$:

$$|g^\star(x_1, a, x') - g^\star(x_2, a, x')| = \frac{\rho(x')|T(x' \mid x_1, a) - T(x' \mid x_2, a)|}{(T(x' \mid x_1, a) + \rho(x'))(T(x' \mid x_2, a) + \rho(x'))}. \quad (75)$$

We use this to prove the following result:

**Lemma 11.** *With probability at least* $1 - \delta$ *we have:*

$$\Pr_{x_1, x_2 \sim \mu} \left( \phi^\star(x_1) \neq \phi^\star(x_2) \wedge \hat{\phi}(x_1) = \hat{\phi}(x_2) \right) \leq \frac{8|\mathcal{A}^2|\Delta}{\Gamma}.$$

*Proof.* Let's define a shorthand $\mathcal{E} = \mathbf{1}\{\hat{\phi}(x_1) = \hat{\phi}(x_2)\}$. Starting with left hand side of Lemma 10 we get:

$$\mathrm{E}_{(x_1, x_2, a, x') \sim D_c} \left[ \mathcal{E} | g^\star(x_1, a, x') - g^\star(x_2, a, x') | \right]$$

$$= \mathrm{E}_{(x_1, x_2, a, x') \sim D_c} \left[ \mathcal{E} \frac{\rho(x')|T(x' \mid x_1, a) - T(x' \mid x_2, a)|}{(T(x' \mid x_1, a) + \rho(x'))(T(x' \mid x_2, a) + \rho(x'))} \right]$$

$$= \sum_{x_1, x_2, a, x'} \frac{\mathcal{E}}{|\mathcal{A}|} \frac{D(x_1 \mid x', a)\rho(x')}{(T(x' \mid x_1, a) + \rho(x'))} \frac{D(x_2 \mid a, x')\rho(x')}{(T(x' \mid x_2, a) + \rho(x'))} |T(x' \mid x_1, a) - T(x' \mid x_2, a)|$$

$$\geq \sum_{x_1, x_2, a, x'} \mathcal{E} \frac{\mu(x_1)\mu(x_2)}{4|\mathcal{A}|^3} |T(x' \mid x_1, a) - T(x' \mid x_2, a)|, \quad \text{using Equation 74}$$

$$\geq \sum_{x_1, x_2, a} \mathbf{1}\{\phi^\star(x_1) \neq \phi^\star(x_2)\} \mathcal{E} \frac{\mu(x_1)\mu(x_2)}{2|\mathcal{A}|^3} \Gamma$$

$$= \frac{\Gamma}{2|\mathcal{A}|^2} \Pr_{x_1, x_2 \sim \mu} \left( \phi^\star(x_1) \neq \phi^\star(x_2) \wedge \hat{\phi}(x_1) = \hat{\phi}(x_2) \right)$$

where the last inequality uses $\frac{1}{2} \sum_{x'} |T(x' \mid x_1, a) - T(x' \mid x_2, a)| = \|T(\cdot \mid x_1, a) - T(\cdot \mid x_2, a)\|_{\mathrm{TV}}$ which is either zero when $\phi^\star(x_1) = \phi^\star(x_2)$ or at least $\Gamma$. Combining these two conditions we get a lower bound of $\mathbf{1}\{\phi^\star(x_1) \neq \phi^\star(x_2)\}\Gamma$ on TV distance. The proof is then completed with application of Lemma 10. □

**Theorem 6.** *(Initial State Clustering Result). Let* $N > 1$ *and let* $\Delta < \frac{\mu_{min}^2 \Gamma}{32N^2|\mathcal{A}|^2} \left( 1 - \left( 1 - \frac{2\mu_{min}}{N} \right)^2 \right)$. *Then there exists a bijection mapping* $\sigma : \mathcal{S}_1 \to [N]$ *such that with probability at least* $1 - \delta$:

$$\forall s \in \mathcal{S}_1, \quad \Pr_{x \sim \mu} \left( \phi^\star(x) = s \mid \hat{\phi}(x) = \sigma(s) \right)$$

$$> 1 - \frac{16N^2|\mathcal{A}|^2\Delta}{\Gamma\mu_{min}^2} = 1 - \frac{16N^2|\mathcal{A}|^2}{\Gamma\mu_{min}^2} \sqrt{\frac{C(\mathcal{F} \circ \Phi)\ln(1/\delta)}{n}}. \quad (76)$$

*Proof.* We will use $i \in [N]$ to denote a decoder state defined by $\{x \mid \hat{\phi}(x) = i, x \in \mathcal{X}_1\}$ and $s \in \mathcal{S}_1$ to denote a real state defined by $\{x \mid \phi^\star(x) = s, x \in \mathcal{X}_1\}$. For any $i, s$ we can bound the left hand side of Lemma 11 as:

$$\Pr(\phi^\star(x_1) \neq \phi^\star(x_2) \wedge \hat{\phi}(x_1) = \hat{\phi}(x_2)) \quad (77)$$

$$= \Pr(\cup_{j \in [N], \tilde{s} \in \mathcal{S}_1} \phi^\star(x_1) = \tilde{s}, \phi^\star(x_2) \neq \tilde{s}, \hat{\phi}(x_1) = j, \hat{\phi}(x_2) = j) \quad (78)$$

$$\geq \Pr(\phi^\star(x_1) = s, \phi^\star(x_2) \neq s, \hat{\phi}(x_1) = i, \hat{\phi}(x_2) = i) \quad (79)$$

$$= \Pr(\phi^\star(x_1) = s, \hat{\phi}(x_1) = i) \Pr(\phi^\star(x_2) \neq s, \hat{\phi}(x_2) = i) \quad (80)$$

$$= \Pr(\phi^\star(x) = s, \hat{\phi}(x) = i) \left\{ \Pr(\hat{\phi}(x) = i) - \Pr(\phi^\star(x) = s, \hat{\phi}(x) = i) \right\} \quad (81)$$

where the second last step follows from observing that $x_1$ and $x_2$ are sampled independently. We define a few shorthands: $\Pr(i) = \Pr(\hat{\phi}(x) = i)$, $\Pr(s) = \Pr(\phi^\star(x) = s) = \mu(s)$ and $\Pr(i, s) = \Pr(\phi^\star(x) = s, \hat{\phi}(x) = i)$. This combined with above and Lemma 11 gives us:

$$\forall i \in [N], s \in \mathcal{S}_1, \quad \Pr(i, s) \left( \Pr(i) - \Pr(i, s) \right) \leq \Delta' := \frac{8|\mathcal{A}|^2\Delta}{\Gamma} \quad (82)$$

We define a mapping $\sigma : \mathcal{S}_1 \to [N]$ as follows:

$$\sigma(s) = \arg\max_{i \in [N]} \Pr(i, s) \tag{83}$$

This gives us:

$$\Pr(\sigma(s), s)\left(\Pr(\sigma(s)) - \Pr(\sigma(s), s)\right) \leq \Delta' \tag{84}$$

Since $\Delta'$ can be brought arbitrarily small, we will assume $\Delta' < {\Pr(\sigma(s))^2}/{4}$ which allows us to write:

$$\Pr(\sigma(s), s) > \frac{\Pr(\sigma(s)) + \sqrt{\Pr(\sigma(s))^2 - 4\Delta'}}{2}, \text{ or} \tag{85}$$

$$\Pr(\sigma(s), s) < \frac{\Pr(\sigma(s)) - \sqrt{\Pr(\sigma(s))^2 - 4\Delta'}}{2} \tag{86}$$

By definition of $\sigma(s)$ we have:

$$\Pr(\sigma(s), s) \geq \frac{1}{N}\sum_{i=1}^{N}\Pr(i, s) = \frac{\Pr(s)}{N} \geq \frac{\mu_{\min}}{N}, \tag{87}$$

where the first inequality uses the fact that maximum of a set of values is greater than its average, and the last inequality uses Assumption 4. We now place another condition on $\Delta'$, namely,

$$\Delta' < \frac{\Pr(\sigma(s))^2}{4}\left(1 - \left(1 - \frac{2\mu_{\min}}{\Pr(\sigma(s))N}\right)^2\right), \tag{88}$$

which implies that:

$$\Pr(\sigma(s), s) < \frac{\Pr(\sigma(s)) - \sqrt{\Pr(\sigma(s))^2 - 4\Delta'}}{2} < \frac{\mu_{\min}}{N}. \tag{89}$$

This eliminates Equation 86. Hence the $\Pr(\sigma(s), s)$ must satisfy Equation 85 which can be simplified as:

$$\Pr(\sigma(s), s) > \frac{\Pr(\sigma(s)) + \sqrt{\Pr(\sigma(s))^2 - 4\Delta'}}{2} \tag{90}$$

$$= \frac{\Pr(\sigma(s))}{2}\left(1 + \left(1 - \frac{4\Delta'}{\Pr(\sigma(s))^2}\right)^{\frac{1}{2}}\right) \tag{91}$$

$$\geq \Pr(\sigma(s))\left(1 - \frac{2\Delta'}{\Pr(\sigma(s))^2}\right) \tag{92}$$

$$\geq \Pr(\sigma(s))\left(1 - \frac{2N^2\Delta'}{\mu_{\min}^2}\right) \tag{93}$$

where the third step uses $\sqrt{1 - y} \geq 1 - y$ for $y \in [0, 1]$ and that ${4\Delta'}/{\Pr(\sigma(s))^2} < 1$ from constraints on $\Delta'$, and the last step uses $\Pr(\sigma(s)) \geq \Pr(\sigma(s), s) \geq {\mu_{\min}}/{N}$ (Equation 87). We can finally prove our main result as:

$$\Pr(s \mid \sigma(s)) = \frac{\Pr(s, \sigma(s))}{\Pr(\sigma(s))} \geq 1 - \frac{2N^2\Delta'}{\mu_{\min}^2} = 1 - \frac{16N^2|\mathcal{A}|^2\Delta}{\Gamma\mu_{\min}^2}.$$

What is left is to show that $\sigma(s)$ is a bijection mapping and collect all constraints on $\Delta'$. Let $s$ and $s'$ be two initial states such that $\sigma(s) = \sigma(s') = k$. We then get:

$$\Pr(k, s)\Pr(k, s') \leq \Pr(k, s)\left(\Pr(k) - \Pr(k, s)\right) \leq \Delta' \tag{94}$$

where the last equality follows from Equation 84. Further, we have $\Pr(k, s) \geq {\mu_{\min}}/{N}$ and $\Pr(k, s') \geq {\mu_{\min}}/{N}$ from Equation 87. This gives us $\frac{\mu_{\min}^2}{N^2} \leq \Delta'$. Hence, if $\Delta' < \frac{\mu_{\min}^2}{N^2}$, then, we cannot have two different initial states mapping to the same decoder state. Further, as $|\mathcal{S}_1| = N$, hence the map $\sigma : \mathcal{S}_1 \to [N]$ is a bijection.

Finally, we made three constraints on $\Delta'$. The first is $\Delta' < \Pr(\sigma(s))^2/4$, second is Equation 88, and third is $\Delta' < \frac{\mu_{\min}^2}{N^2}$. Note that Equation 88 already implies that $\Delta' < \Pr(\sigma(s))^2/4$. Hence, the overall constraint on $\Delta'$ is:

$$\Delta' < \min\left\{\frac{\mu_{\min}^2}{N^2}, \frac{\Pr(\sigma(s))^2}{4}\left(1 - \left(1 - \frac{2\mu_{\min}}{\Pr(\sigma(s))N}\right)^2\right)\right\} \tag{95}$$

We can simplify this constraint by making it tighter using $\Pr(\sigma(s)) \in \left[\frac{\mu_{\min}}{N}, 1\right]$:

$$\Delta' < \min\left\{\frac{\mu_{\min}^2}{N^2}, \frac{\mu_{\min}^2}{4N^2}\left(1 - \left(1 - \frac{2\mu_{\min}}{N}\right)^2\right)\right\} = \frac{\mu_{\min}^2}{4N^2}\left(1 - \left(1 - \frac{2\mu_{\min}}{N}\right)^2\right). \tag{96}$$

Note that we are assuming here that $N \geq 2$ and, therefore, $\mu_{\min} < 1$, which implies $2\mu_{\min}/N \leq \mu_{\min} < 1$. When $N = 1$, we can trivially align the single initial state. This completes the proof. $\qquad\square$

This allows us to separate all initial states at time step $h = 1$ with high probability.

## C.2   Aligning the learned decoder states

The only thing left is aligning learned decoder states in the target domain with simulator initial states, which we do as follows.

At a high level Algorithm 3 first clusters the initial observation into clusters, then each time it start with a cluster, it run a subroutine, InitialStateTest, to test the latent state index of that cluster. The testing algorithm Algorithm 4, run our main algorithm `TASID` the hypothesis of the state index from $0$ to $N$. By the analysis of `TASID`, we know that if the hypothesis is correct, it will learn a policy with nearly robust value. Thus for each cluster, with in $Nn$ episodes the algorithm will find the nearly robust policy.

**Theorem 7.** *If* $\epsilon_{\mathcal{F}} \leq \frac{(1-2\eta)^2\epsilon}{4H^4|\mathcal{A}|^3}$ *we learn the initial state decoder* $\hat{\phi}$ *with* $n_0$ *samples such that*

$$n_0 \geq \max\left\{\frac{256H^2N^4|\mathcal{A}|^4C(\mathcal{F}\circ\Phi)\ln(1/\delta)}{\epsilon^2\Gamma^2\eta^4}, \frac{1024N^4|\mathcal{A}|^4C(\mathcal{F}\circ\Phi)\ln(1/\delta)}{\Gamma^2\eta^6}\right\},$$

*Algorithm 3 will execute a policy that is close to robust policy by* $5\epsilon$ *in all but*

$$\mathcal{O}\left(\frac{N^2H^4|\mathcal{A}|^3\ln(N^2|\mathcal{F}|/\delta)}{\epsilon(1-2\eta)^2} + \frac{N^2H^2\ln(N^2/\delta)}{\epsilon^2} + \frac{N^4|\mathcal{A}|^4C(\mathcal{F}\circ\Phi)\ln(1/\delta)}{\Gamma^2\eta^2}\max\left\{\frac{H}{\epsilon^2}, \frac{1}{\eta^2}\right\}\right)$$

*episodes with probability at least* $1 - 3\delta$.

*Proof.* We prove this theorem by two steps. First, we show that for each cluster $\hat{\phi}(x)$, after we run Algorithm 4 with $N(n_l + n_t)$ steps we can find the correct state of that cluster. Second, we show that once we find the correct state, we run `TASID` and learn a policy at most $4\epsilon$ worse than the robust policy.

First, by the definition of $n_0$, we have that $\Delta := \sqrt{\frac{C(\mathcal{F}\circ\Phi)}{n_0}\ln\left(\frac{1}{\delta}\right)} \leq \frac{\eta^2\Gamma}{32N^2|\mathcal{A}|^2}\frac{2\eta}{N} < \frac{\eta^2\Gamma}{32N^2|\mathcal{A}|^2}\left(1 - \left(1 - \frac{2\eta}{N}\right)^2\right)$ for $N \geq 2$ and $\eta < 1$. By Theorem 6, we have for each $i \in [N]$, there must exist a state $\sigma(i) \in [N]$ such that with probability at least $1 - \delta$,

$$\Pr(\phi(x) = s|\hat{\phi}(x) = i) \geq 1 - \frac{16N^2|\mathcal{A}|^2\Delta}{\Gamma\eta^2} \tag{97}$$

$$\geq 1 - \frac{\epsilon}{H} \tag{98}$$

Thus if we run Algorithm 4 InitialStateTest for $s = \sigma(i)$. Then by Theorem 3, we have that the value of learned policy $\pi$ is at least

$$V_{\mathbf{M}}^{\rho\circ\phi^\star} - \frac{8H^4|\mathcal{A}|^3\ln(N^2|\mathcal{F}|/\delta)}{n_l(1-2\eta)^2} - \frac{4H^4|\mathcal{A}|^3\epsilon_{\mathcal{F}}}{(1-2\eta)^2} - H\epsilon_0 = V_{\mathbf{M}}^{\rho\circ\phi^\star} - 3\epsilon,$$

with probability at least $1 - \delta/N^2$. By Hoeffding's inequality, we know that the Monte-Carlo estimates $v(s)$ of the learned policy value with $N_t$ samples is at least

$$V_{\mathbf{M}}^{\rho \circ \phi^\star} - 3\epsilon - \sqrt{\frac{H^2 \ln(N^2/\delta)}{2n_t}} = V_{\mathbf{M}}^{\rho \circ \phi^\star} - 4\epsilon,$$

with probability at least $1 - \delta/N^2$. Thus let $\hat{s} = \operatorname{argmax}_s v(s)$ and $\pi$ be the corresponding learned policy from TASID given hypothetical initial state $\hat{s}$. The policy value $v^\pi$ is at least

$$v(\hat{s}) - \epsilon \geq V_{\mathbf{M}}^{\rho \circ \phi^\star} - 5\epsilon \tag{99}$$

with probability $1 - 2\delta/N$ by taking the union bound on all $N$ possible hypothetical initial states. Thus, for a given state cluster, after run Algorithm 4 InitialStateTest for $N(n_l + n_t)$ episodes, the policy value is at least $V_{\mathbf{M}}^{\rho \circ \phi^\star} - 5\epsilon$. That means for each state cluster, we make at most $N(n_l + n_t)$ mistakes. Thus we make at most $N^2(n_l + n_t)$ mistakes in total during running Algorithm 4. Notice that we also need $n_0$ samples to learn the decoder $\hat{\phi}$. The total number of episodes we may make mistake on is therefore

$$N^2(n_l + n_t) + n_0$$

Taking the union bound over all initial state clusters $\hat{\phi}(x)$ and the high probability statement in Theorem 6, we have that with probability $1 - 3\delta$ the statement holds. We finish the proof by plugging in $n_l, n_t, n_0$. $\square$

## D  Experiment Details

### D.1  Experiment Details in Combination Lock

**Domain details**  We describe the details of the combination lock domain here. The deterministic MDP is described in section 5. The transition dynamics and reward functions in the target domain is defined by the the coefficients $\xi(a'|s, a)$ and Definition 1. For each $s, a$, we set $\xi(a'|s, a) := \eta p(a'|s) + (1 - \eta) \mathbb{1}(a' = a)$ where $p(\cdot|s)$ is a random probability mass distribution and each probability is drawn uniformly from $[0, 1]$ and then normalized. We set $\eta = 0.1$ in the experiment. The only exception is that all transitions from state $(H, 2)$ are not perturbed in the true target domain. This settings makes the robust policy without this knowledge not optimal in this specific instance, but our goal is still learning the robust policy here.

The observation mapping is defined as below. Let $v_s$ be a 2-sparse encoding in $\mathbb{R}^{H+4}$ of the state $(h, i)$ where the first 3 bits is a 1-sparse encoding of $i$ and last $H + 1$ bits is a 1-sparse encoding of $h$. The the observation $o$ is computed by

$$o = \mathbf{H} \times \operatorname{perm}(v_s + \mathcal{N}(0, 0.01)), \tag{100}$$

where perm is an arbitrary permutation of $[H + 4]$, and $\mathbf{H}$ is the $2^{\lceil \log_2(H+4) \rceil}$ by $2^{\lceil \log_2(H+4) \rceil}$ Hadamard matrix, consists of $2^{\lceil \log_2(H+4) \rceil}$ mutually orthogonal columns. The permutation $\operatorname{perm}(v)$ operator shuffles the dimensions of the vector $v$. E.g., if perm denotes the permutation $(3, 1, 2)$, then $\operatorname{perm}(v) = (v_3, v_1, v_2)$. We create a Hadamard matrix $H_n$ of size $n \times n$ for $n = 2^l$ for some $l \in \mathbb{N} \cup \{0\}$ using Sylvester's construction.[4] If $H + 4 < 2^{\lceil \log_2(H+4) \rceil}$, then the vector $\operatorname{perm}(v_s + \mathcal{N}(0, 0.01))$ will be padded with zeros to ensure it is of size $2^{\lceil \log_2(H+4) \rceil}$. Our constructions are significantly based on previous hard combination locks studied in Du et al. [2019], Misra et al. [2020].

**Implementation details of TASID**  We describe the details of TASID and hyper-parameters below. We model the action predictor class $\mathcal{F}$ by a two-layer MultiLayer Perceptron (MLP) with Leaky ReLU activations in this domain. The input to the MLP is the concatenation of observations $x$ and $x'$. This is processed through two non-linear layers, and finally a softmax operation is performed to generate probabilities over actions. We implement the model in PyTorch and train it using Adam optimization.

---

[4]Sylvester's construction defines $H_1 = [1]$ and $H_{2k} = \begin{bmatrix} H_k & H_k \\ H_k & -H_k \end{bmatrix}$ for all $k \in \mathbb{N}$.

The $h + 1$-th action predictor, i.e., $\hat{f}_{h+1}$ is initialized with the parameter from $h$-th action predictor ($f_h$). We initialize the first action predictor ($f_1$) using PyTorch's default initialization. For action predictor we remove 20% of the $n_D$ datapoints for use as a held-out validation set and use the remaining for training parameters of the MLP. We perform 100 epochs of training and evaluate the current model on the validation set after each epoch. We stop training if the validation loss does not increase for 10 consecutive epochs. We save the model after each epoch, and use the model with the smallest validation loss. The only hyperparameter we search for `TASID` is $n_D$ whose values we search over the grid $\{1000, 2500, 10000\}$.

For all algorithms we report results with best hyperparameter for every value of $H$. For every value of $H$ and hyperparameter setting, we run the experiment 5 times with different seeds. We select the best hyperparameter as one that takes the least median number of episodes to achieve a moving average return of $0.95v^\rho$, where $\rho$ is the latent robust policy and $v^\rho$ is its value in the target domain. We compute median across 5 seeds. In case of a tie, we look at the average return over the entire course of training and across all seeds.

The hyper-parameters of training neural networks is the same across our methods and all baseline methods, and are specified in Table 2 on page 31.

| parameter name | values |
|---|---|
| hidden dimension | 56 |
| learning rate | 0.0003 |
| optimizer | Adam |
| batch size | 32 |
| gradient clipping | 0.25 |

Table 2: Hyperparameters in training NNs for all algorithms in the combination lock domain

**Implementation details of PPO, PPO+RND, and domain randomization**   We train PPO and PPO+RND, with a maximum of $5 \times 10^5$ number of episodes in the target domain. The policy network is implemented by a two-layer MLP with ReLU activations in this domain, and the training hyperparameters are the same as Table 2. When training RND, the observation is normalized to mean 0 and standard deviation 1. Hyperparameters that are specific to the PPO and RND are listed in Table 3.

We perform a grid search over the entropy coefficient in PPO and the coefficient of RND bonus. The grid search we use for PPO+RND experiments contains 15 different hyperparameter choices (5 for RND bonus and 3 for entropy coefficient). This results in $15 \times 5 = 75$ experiments for every value of $H$. In contrast, the grid search for `TASID` only contains 3 different hyperparameter choices. To avoid further increasing the search space, we only use the best PPO+RND hyperparameters for every value of $H$ when running the domain randomization experiments. The best hyperparameter values are chosen using the procedure described earlier with sole exception we use compute episodes needed to achieve a moving average return of $0.5v^\rho$ instead of $0.95v^\rho$. This weaker metric is more helpful since the latter often gives infinities.

For domain randomization pretraining, we first train in the source domain with $5 \times 10^5$ episodes. During this training, every 100 episodes, we randomize the domain uniformly by regenerating a permutation function and the perturbation in transitions $\xi(\cdot|\cdot, \cdot)$, in the same way as how the target domain is generated. We assume that the domain randomization algorithm knows $\eta$ (same as `TASID`), and everything about the observation mapping except the specific permutation function and transition perturbation (not known by `TASID`).

Rather than uniform domain randomization, EPOpt [Rajeswaran et al., 2017] uses only episodes with a return smaller than lower $\epsilon$-quantile in the batch (100 episodes) during domain randomization pretraining. We searched for values of $\epsilon$ in $\{0.1, 0.2, 0.5, 1.0\}$ where $1.0$ gives uniform domain randomization. Therefore, the hyperparameter grid search for domain randomization has 4 different choices.

**Details of the experiment results**   We report the details of experiments we used to generate Figure 2b in the paper. We run each algorithm with 5 random seeds for a maximum of $5 \times 10^5$ number

| parameter name | values |
|---|---|
| clip epsilon | 0.1 |
| discounting factor | 0.999 |
| number of updates per batch | 4 |
| number of episodes per batch | 100 |
| entropy loss coefficient | $0, 0.01, 0.001$ |
| RND bonus coefficient | $0, 100, 500, 1000, 10000$ |
| $\epsilon$ in EPOpt | $0.1, 0.2, 0.5, 1.0$ |

Table 3: Hyperparameters for PPO, PPO+RND, PPO(+RND) with domain randomization in the combination lock domain. Multiple values stand for the values we run grid-search over, and we report the best performance among them for each environment specification (each $H$).

of episodes in the target domain, saving checkpoints every 1000 episodes. We decide whether the combination lock is solved or not if the algorithm can achieve a moving average return of $0.95v^\rho$. In Figure 2b we report the median of the number of episodes needed to solve the combination lock with horizon $H$. The results of each random seeds is list in the following table.

| Algorithm | Horizon | Number of episodes needed |
|---|---|---|
| TASID | 5 | 17000, 241000, 9000, 175000, 9000 |
| | 10 | 29000, 32000, 33000, 41000, 30000 |
| | 15 | 42000, 43000, 41000, 55000, 42000 |
| | 20 | 53000, 56000, 54000, 52000, 63000 |
| | 25 | 341000, 142000, 67000, 198000, 69000 |
| | 40 | 123000, 217000, 303000, 156000, 138000 |
| PPO | 5 | 69000, 128000, 58000, 31000, 81000 |
| | $\geq 10$ | $\infty, \infty, \infty, \infty, \infty$ |
| PPO+RND | 5 | 41000, 18000, 12000, 17000, 27000 |
| | 10 | 141000, 152000, 103000, 245000, 106000 |
| | 15 | 102000, 318000, 167000, 147000, 111000 |
| | 20 | $459000, 384000, \infty, \infty, 349000$ |
| | 25 | $\infty, 383000, \infty, 458000, \infty$ |
| | 40 | $\infty, \infty, \infty, \infty, \infty$ |
| PPO+DR | 5 | 91000, 77000, 53000, 186000, 66000 |
| | $\geq 10$ | $\infty, \infty, \infty, \infty, \infty$ |
| PPO+RND+DR | 5 | 53000, 40000, 50000, 66000, 143000 |
| | 10 | 170000, 364000, 194000, 252000, 214000 |
| | 15 | $\infty, \infty, \infty, \infty, \infty$ |
| | 20 | $265000, \infty, \infty, \infty, \infty$ |
| | 25 | $409000, \infty, \infty, 352000, \infty$ |
| | 40 | $\infty, \infty, \infty, \infty, \infty$ |

Table 4: Results for all random seeds to generate Figure 2b. A value of $\infty$ denotes a timeout indicating that the algorithm was not able to achieve a moving average return of $0.95v^\rho$ in the maximum allowed number of episodes of $5 \times 10^5$.

In the main paper, Figure 2c shows the reward curves for different algorithms with horizon equals 40 in the target environment. Here we include the reward curves for all values of horizon in Figure 4. It shows our algorithm stably learned a robust policy with number of samples that is smaller than baseline algorithms. Though baseline algorithms aim to learn the near optimal policy, they did not converge to a policy that is significantly better than the robust policy.

**Amount of compute** We run our experiments on a cluster containing mixture of P40, P100, and V100 GPUs. Each experiment runs on a single GPU in a docker container. We use Python3 and

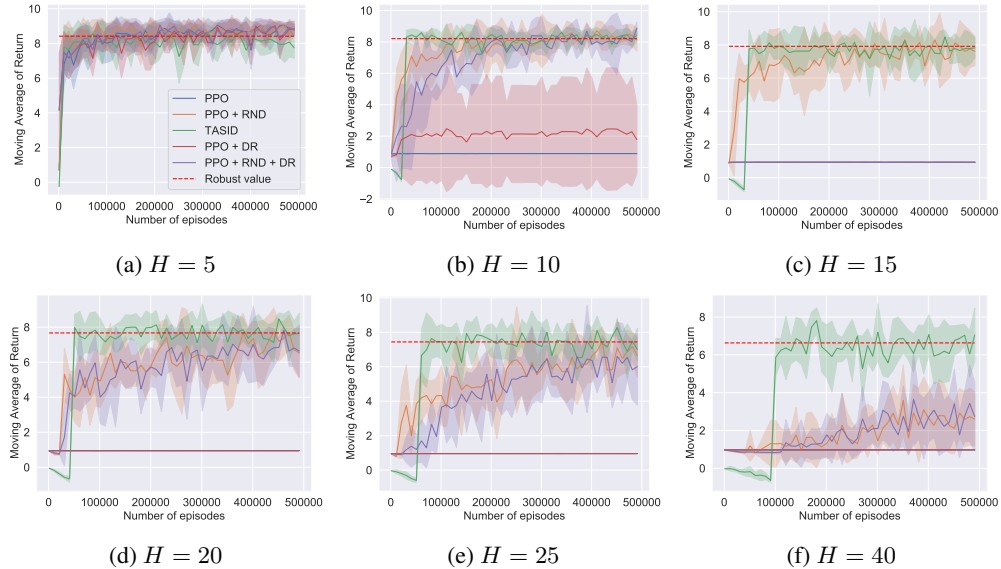

Figure 4: Reward curves in target environments with different horizons. The legend is shown in the plot for $H = 5$, and is shared with all other plots.

Pytorch 1.4. We found that for H=40 and V100 GPUs, on average, POTAS took 2.5 hours, PPO + DR took 14 hours, and PPO + RND + DR took 31 hours. Performing domain randomization increased the computational time by a factor of 2.

## D.2 Experiment Details in MiniGrid

**Domain details** We use the code of the MiniGrid environment [Chevalier-Boisvert et al., 2018] under the Apache License 2.0. In MiniGrid, the agent is placed in a discrete grid world and needs to solve different types of tasks. We customized the lava crossing environment in several ways. We first create a shorter but more narrow crossing path, as the optimal path in deterministic environment. Then we construct a longer but more safe path under perturbation. The map of the mini-grid is shown in Figure 3a. The height of the map can be changed without changing the problem structure and the optimal and robust policy. The state consists of the $x$-$y$ coordinate of the position, a direction that takes four values, and the time step. The agent only observe a $7 \times 7$ area in front of it. (see Chevalier-Boisvert et al. [2018] for details.) We set the observation to be the visual map (RGB image) with a random noise between $(50, 50, 50)$ and $(150, 150, 150)$ for all background pixels. The action space is changed to having five actions: moving forward, turning left, turning right, turning right and moving forward, turning left and moving forward. We set the horizon to three times the number of steps the robust policy needs to reach the goal in the deterministic environment.

**Implementation details of** `TASID` We describe details of `TASID` and hyper-parameters below. We implement the action predictor model class by a convolutional neural network (CNN). We take the two images $x$ and $x'$ and concatenate them along the channel dimension. This concatenated image is passed through CNN which applies two convolutional layers with ReLu activations followed by a flattening the feature and applying a linear layer to reduce it to a vector of size the number of actions. Finally, we apply a softmax layer to generate probabilities over actions. We train the algorithm with $5 \times 10^5$ episodes, for all values of $\eta$ and gridworld height. We run each experiment 5 times and report averaged results in Figure 3. Shaded bands in the Figure correspond to the $95\%$ Student's $t$ confidence intervals. Other hyperparameters for minigridworld experiments are shown in Table 5.

**Amount of compute** We run our experiments on a cluster containing P100 GPUs. Each experiment runs on a single GPU in a docker container. We use Python3 and Pytorch 1.4. We found that on P100 GPUs, POTAS took 2 to 6 hours (2 hours for $\eta = 0.1$ and 6 hours for $\eta = 0.5$).

| parameter name | values |
| --- | --- |
| CNN kernel 1 | $8 \times 8 \times 16$, stride 4 |
| CNN kernel 2 | $4 \times 4 \times 32$, stride 2 |
| learning rate | 0.0003 |
| optimizer | Adam |
| batch size | 256 |
| gradient clipping | 100 |

Table 5: Hyperparameters for `TASID` in the MiniGrid domain