# OpenReview forum: "Provably sample-efficient RL with side information about latent dynamics"
_NeurIPS.cc/2022/Conference — NeurIPS 2022 Accept_

### Official Review · Reviewer_Jw57 · 2022-06-24

**Rating:** 7
**Confidence:** 3
**Soundness:** 3 good
**Presentation:** 3 good
**Contribution:** 3 good

**Summary:**

The authors present TASID, an algorithm which leverages knowledge of an abstract MDP (through an abstract simulator) to efficiently solve a more difficult MDP, using standard assumptions in the block MDP framework. They illustrate the benefits of their method in a synthetic toy problem and a slightly more challenging MiniGrid environment.

**Questions:**

In Section 2.1, two things could be done to help the reader better understand the level of difficulty of the approach.
- from one side, it must be made clear why solving block MDPs is less daunting than solving full POMDPs. If I understoos correctly the literature, block MDPs are "one-step decodable POMDPs", so a memory of size one is enough. Still if I'm correct, this is why learning an inverse dynamics model with x_h and x_{h+1} is enough to efficiently decode the latent state. I think making this relationship to POMDPs would help non-expert readers like me.
- from the other side,  it must be made clear why solving block MDPs is not trivial. From the current description one may understand that, since two states cannot emit the same observation, one can trivially recover the state from the observation. I think it should be made clear that the "perfect decoder" needs some memory (actually, one-step memory if I'm correct).

Another unclear point is related to exploration and the difficulties it raises in the framework. In particular, lines 268-269 state "We theoretically showed that TASID can solve problems that require performing strategic exploration using a small number of episodes." but the word exploration does not appear at all in the methods. So somehting is not explicit enough here. More precisely, it is unclear to me whether considerations on the perturbation level \eta is related to exploration or more to environment noise.


**Limitations:**

Does not apply

**Strengths And Weaknesses:**

Strengths:
- the paper addresses a relevant problem, for instance when a robot but navigate a complex environment but is given a simplified map.
- it looks technically sound (I'm not expert enough of the domain to spot potential flaws)

Weaknesses:
- although the target application is complex robotics navigation problems, the method is demonstrated with toy environments that are not representative of the targeted complexity. We would like to see something closer to the "running example" put forward at the beginning of the problem statement...
- being naive on block MDPs, I had to read other papers to better understand the relationship to POMDPs. A few points could be clarified, see below.
- rejecting part of the related work in appendices is not a good practice. I understand that it is difficult to fit this paper into 9 pages. Maybe the authors could rather move Algorithm 2 to the appendix?

---

> ### Author Response · Authors · 2022-08-02
> **Author Response: Clarification on Block MDP**
>
> We thank you for the feedback. We address the different points below:
>
> 1. **Non-triviality of Block MDP:** One reason why decoding state in Block MDP is non-trivial is that the set of observations that a state can emit can be infinitely large. In practice, this can happen due to image/sensor noise in a sensor-like camera. This requires the agent to learn a decoder that is accurate for distribution over an infinite set of observations when given access to a small set of observational data.
>
> 2. **Related work:** We did our best to mention the most relevant related work in the introduction--partly via inline citations and some additional work in a separate paragraph. We then added a more in-depth discussion of less relevant work in the Appendix to do justice to the much larger literature. Please let us know which additional references we should highlight in the main paper.
>
> 3. **Block MDP vs POMDP:** Block MDPs are a subset of POMDPs where the current state can be decoded from just the current observation. A typical example will be a robot navigating in a room with an overhead or a panoramic camera that ensures that if the robot is at two different spots in the room, then the camera does not generate the same image.

---

> > ### Comment · Reviewer_Jw57 · 2022-08-03
> > **More accurate answers please**
> >
> > Among the reviewers, I'm probably the least expert of the domain (I don't know why I got this paper). So I understand that you may pay more attention to the other reviews. Anyways, I have the feeling that your response does not truly answer some of my questions, which is quite frustrating...
> > So:
> > - can you be more precise on the compared complexity of solving block MDP wrt POMDPs, and explain why it is so much easier?
> > - can you explain how you deal with exploration in your framework?
> >
> > Thanks in advance

---

> > > ### Author Response · Authors · 2022-08-05
> > > **More explanation on Block MDP vs POMDP, and Avoiding Exploration**
> > >
> > > We are happy to describe things in more detail. Please see our response below, and let us know where you want us to explain further.
> > >
> > > ### **Block MDP vs POMDP**
> > >
> > > POMDP is a very general class of problems where the current state is not always decodable from the current observation. In contrast, for Block MDP the current observation is always sufficient to decode the current state.
> > >  It is known that, in general, POMDPs cannot be solved with a polynomial sample complexity. For example, see the lower bound examples for Proposition 1 and Proposition 2 in _“PAC Reinforcement Learning with Rich Observations”_, Krishnamurthy et al. NeurIPS 2016. To make it more intuitive, consider a deterministic world where an agent is navigating blindly, i.e., the agent always receives the same observation. The agent can never decode the current state from observation. It cannot tell if two action sequences lead to the same state or different states. This agent will then have to try all exponential possible action sequences to explore the state space and optimize the reward (any action sequence that it avoids might lead to the goal state where all the reward is).
> > >
> > > In contrast, algorithms with polynomial sample complexity exist for Block MDP. E.g., the algorithms of Du et al., 2019 [1] and Misra et al., 2020 [2] learn a state decoder to map the current observation to the current state. This is done by solving a representation learning task. Note that such a decoder is not possible for POMDPs. Consider our previous example, but this time imagine the agent has a camera and the setting is Block MDP. If one has access to a state decoder, one can tell whether the two action sequences lead to the same state or different states (by decoding the final observation and checking if it maps to the same state for both sequences). This allows us to explore the Block MDP using $|S|$ different action sequences, one sequence for every state, since we can remove redundant paths that visit the same state. In contrast, in our earlier POMDP example, we consider all $A^H$ action sequences.
> > >
> > > ### **Avoiding Exploration**
> > >
> > > If one does not have access to a simulator or prior knowledge, then one must perform strategic exploration (i.e., visit each state with high probability) to find the optimal policy. This is because any state that we don't visit, could be the goal state where all the reward is. This intuitively explains why sample complexity (number of samples needed to learn a near-optimal policy) scales with the size of the state space. It is because the more states there are, the more we need to explore, and each state may require a different policy to optimally visit it. Therefore, the number of policies needed to explore the state space grows with the number of states. And since each policy needs to be executed at least once, therefore, we need samples that grow with the size of state space.
> > >
> > > We avoid this in our setting because we have access to a simulator from which we extract a latent policy, namely the robust policy, that we want to follow. Therefore, instead of optimally visiting each state using a policy for that state, we try to imitate a **single** latent policy. For this imitation task, our algorithm allows us to decode the current observation (drawn from a distribution generated by this single latent policy, environmental noise, and one-step random exploration) to the correct state. All this while, we sample from a single policy, whereas Block MDP algorithms [1, 2] sample from a collection of policies that visit all reachable states. In our analysis, this allows us to avoid dependence on the size of state space.
> > >
> > > One subtle nontriviality is that naively applying the decoding strategies of Block MDP algorithms [1, 2] fails in our setting and either leads to dependence on the size of state space, or an exponential sample complexity in the horizon. Our novel decoding strategy is able to avoid this by carefully utilizing the simulator and solving the right representation learning problem.
> > >
> > > **References**
> > > 1. _Provably efficient RL with Rich Observations via Latent State Decoding_, Du et al., ICML 2019
> > > 2. _Kinematic State Abstraction and Provably Efficient Rich-Observation Reinforcement Learning_, Misra et al., ICML 2020

---

> > > > ### Comment · Reviewer_Jw57 · 2022-08-06
> > > > **Thank you**
> > > >
> > > > I thank the authors for these additional explanations, which were very helpful

---

### Official Review · Reviewer_3VEr · 2022-07-06

**Rating:** 6
**Confidence:** 3
**Soundness:** 2 fair
**Presentation:** 2 fair
**Contribution:** 3 good

**Summary:**

Paper 4378 proposes an algorithm to solve Block MDP when imperfect prior knowledge about the dynamics is available.
They encode this information in a deterministic abstract simulator, assuming the target environment to be a perturbation of it.
The optimal policy in the abstract simulator is used in the target environment with an inverse dynamic model to decode observations to latent states.
The authors provide a PAC analysis of the algorithm's sample complexity which does not depend on the size of the state and observation space.
The algorithm is tested in two environments to evaluate performance and scalability.

**Questions:**

- Mainly addressing the doubts stated in the *Quality* paragraph.
- The sample complexity considers only the interaction with the target environment. However, what is the impact of the robust dynamic programming algorithm used to learn a policy in the abstract simulator in the overall bound? Is it still independent of the size of the state space?
- The experiment on the scalability of TASID is defined only with respect to $|S|$. However, in line $343 you claim independence on the size of the observation space. Could you provide empirical evidence of it, for example running the Visual Grid task with larger observations?

**Limitations:**

- It would be useful to extend the discussion on how the assumptions of a deterministic abstract simulator and perturbed target environment are realistic.

- Improvement on reproducibility:
	- Include instruction on installation requirements. I set up the environment by my own, looking at the missing dependencies and appropriate versions. A simple file "requirements.txt" would make the installation straightforward.
	- Include instructions on how to run the code. A readme file with a few commands to reproduce the main experiments is essential. In the current status, I run the scripts in the "experiments" directory without any knowledge of any input parameters.
	- Include information on expected computing time. Comments on the expected time (and associated platform) to complete each experiment is helpful. For example, running the mini-grid experiments does not show any progress after 15 minutes of execution. Since there is no further information on the current status of the run, I killed the process.
	- Include parameter configuration. Use configuration files to exactly reproduce each experiment.

**Strengths And Weaknesses:**

### Main comment
- *Significance*: The analysis of sample complexity is interesting and the design of algorithms with guaranteed sample complexity is a highly-relevant topic for the RL community. This result relies on the assumption of a deterministic abstract simulator and perturbed target environment which is interesting as well. However, it is unclear if this assumption is realistic enough and its feasibility in more complex problems.
- *Clarity*: The paper is well written and organized, with the exception of the related work section that is too short. An extended version of it is included in the appendix. Moreover, some parts of the proofs are not clear, such as the definitions of the various probability distributions at the beginning of Algorithm 2's proof, line 577.

- *Originality*: The main criticism of this work is about novelty. In fact, it builds on top of prior works (adequately cited in the related works) and uses standard techniques, such as optimizing the policy in the abstract simulator and learning a decoder to map observations to latent states. The proof also adapts existing results (Agarwal 2020) to this work.
- *Quality*: There are some flaws and inaccuracies that make it difficult to follow some of the proofs and raise doubts about the validity of the overall claims. In particular:
	- It is not clear what are the assumptions on the function class $F$ which approximates the decoder, and if the bound holds for finite or infinite hypothesis classes.
	- In the proof of Lemma 1 (Appendix, line 497) ends with a disequality. However, in the following line (498), equivalence is stated and it is not clear why.
	- In the proof of Algorithm 2 (Appendix, line 577), the distribution $P_h(x)$ is defined referring to Algorithm 2, Line 9. However, these definitions are not formally defined.
	- In the proof of Algorithm 2 (Appendix, lines 576 and 582), the distribution $Unf(A)$ is first defined as a uniform distribution over action space, and later as uniform conditional distribution of $a$ given $x$.

### Other remarks
- *Reproducibility*: The authors provide an appendix and code as supplementary material. However, the code presents several flaws and requires a certain effort from the readers to reproduce the results. I have been able to run the code for the combination-lock, while the mini-grid gets stuck in the first iteration. It could depend on computation time but no further information is provided in the log, so it is difficult to judge. Then, I have not been able to reproduce the results. I will include hints to improve on that later on.

---

> ### Author Response · Authors · 2022-08-02
> **Author Response: Major clarification on dynamic programming, and confirming the correctness of code**
>
> Thank you for your detailed feedback.  We address all the remarks in the review starting with an important clarification.
>
> 1. **Major clarification:** In robust dynamic programming, the simulator is already given to the agent, therefore, the sample cost of robust dynamic programming is 0 (as there is no interaction with an environment). There is, however, a computational cost and it will be linear in the number of states. We can employ heuristics such as approximate dynamic programming techniques to further reduce the computational complexity, however, this is not the focus of our study.
>
> 2. **Significance:** Our main motivation behind _"near-deterministic" setting_ stems from its application in robotics, where such assumptions are relatively common and which is also a crucial application area for sim-to-real or transfer RL setting. Our assumptions also capture sticky actions which are common in deep RL settings.
>
>       Certainly, near-deterministic assumptions are inapplicable to certain problems, and this restriction is not made for general block MDP settings (Du et al., 2019, Misra et al., 2020), however, these algorithms have sample complexity that depends on the size of the state space, a dependence that we crucially remove. Studying whether one can avoid dependence on state space while transferring from a stochastic simulator, is an interesting future work direction. However, we believe that our work contains novel and interesting results with an interesting set of applications and, therefore, stands on its own.
>
> 3. **Originality:** We note that our work is the **first algorithm** that achieves target environment sample complexity in our setting which is independent of the size of state space. Our work employs a novel decoding approach that learns a non-Markovian decoder by relying on learning an inverse dynamics model and using it together with the simulator.
>
>
>       We also note that the result that we use from Agarwal 2020, is a classic result on generalization guarantee for maximum likelihood that is well-known for decades (e.g., Van de Geer, 2000, Chapter 7). Further, this result is not the main part of the theoretical analysis but we state it as an external result that we use similar to a VC-dimension generalization bound. Our main theoretical analysis is to show that our different pieces (robust dynamic programming, decoding, learning inverse dynamics, use of simulator), fit together and result in learning a robust policy with a polynomial sample complexity that is independent of the size of state space.
>
> 4. **Quality:** Our responses to the reviewer’s questions are listed below:
>
> -   We state this assumption in Assumption 3 (line 240).
>
> -   We apologize for that typo and thank you for pointing it out. The last line in the proof of Lemma 1 right before line 498 should be equality. The equality holds by the definition of the perturbation set and min operator. It does not affect the correctness of the proof.
>
> -   We define $P_h(x)$ in text in algorithm 2, Line 7 as the distribution over observations induced by following the learned policy $\pi_{1:h-1}$ for the first $h-1$ steps. While we think this text definition is clear, we will add a notation table at the start of the Appendix for all commonly used notations to make it more clear.
>
> -   We always use $unf(A)$ as the uniform distribution over action space, in both lines you refer to. We will clarify this but no change in maths is needed.
>
> 4. **Reproducibility:** Thank you for the feedback on the code. We re-ran our submitted code on a different computer and confirm the code’s correctness. We explain the observation on running time for minigrid experiments below, **which we can confirm is not a bug**. We were unable to find any other “flaws” that were referred to in the review.
>
>
>       In minigrid, in each iteration, the first step is to collect the data and this takes time with the minigrid environment which is an external environment library. We print the log of collecting data, learning process, and performance of the learned policy. However, it will not print any log until finishing the data collection part. To run the code faster, one can pass a smaller number of samples (--samples) in the command. We will update the README to clarify this.
>
>
>       We list the specification we use to run our experiment in the appendix and it can take a long time using only CPUs. Further, different environments can take different time depending on the RAM, cores, and library version. We will in the future release a Github repository that will allow us to handle details for a range of systems through issues and wiki.

---

> > ### Comment · Reviewer_3VEr · 2022-08-08
> > **Thank you for clarifying**
> >
> > The authors' explanations were indeed useful to better understand the proposed work.
> >
> > I also appreciate the authors' commitment to releasing an improved code base.
> >
> > After considering the authors' rebuttal, I have positively updated my score.
> > Considering the overall quality and presentation of the work, I do not feel comfortable increasing them further.

---

### Official Review · Reviewer_Wg3g · 2022-07-11

**Rating:** 7
**Confidence:** 3
**Soundness:** 4 excellent
**Presentation:** 4 excellent
**Contribution:** 3 good

**Summary:**

This paper proposes an offline control algorithm for finite-horizon time-inhomogeneous block MDPs. The agent is given observations at each state and an approximate deterministic simulator is available. The agent first infers a nearly-optimal policy the simulator that is robust to perturbations (and so is nearly optimal for the underlying MDP as well) and then, through interaction with the environment, learns a decoder that maps histories of observations to states. Sample complexity of the algorithm is studied and experiments on a synthetic MDP and a gridworld are provided.

**Questions:**

See weaknesses section

**Limitations:**

For horizon H the current algorithm needs H^4 episodes (H^5 steps) but this is stated in the paper. I do not find fixing this limitation within the scope of the current paper.

The current algorithm also has large memory requirements which should be stated.

**Strengths And Weaknesses:**

Strengths:

The paper is clearly written and well organized. It studies a novel setting (to the best of my knowledge) and provides a sound algorithm along with theoretical and empirical results. Altogether I am in favor of acceptance.

The most interesting contribution is the use of eta-perturbation to model the simulator's inaccuracies. Robust dynamic programming with this perturbation and the guarantees in this paper may have other applications in reinforcement learning in the future. The discussion on this perturbation is however limited and it is not clear how strong or practical this measure of perturbation is.

Weaknesses:

1. The paper would greatly benefit from situating its sample complexity against related settings or algorithms. Some questions that such discussion could answer are: (a) If other algorithms have been proposed for this setting, what is their sample complexity? (b) What lower bounds and upper bounds for other types of perturbations and how does eta-perturbation make the problem simpler or harder? (c) If we knew that such deterministic approximate simulator exists for the MDP but it was not available, how would the bounds change?

2. The paper mentions multiple times that the sample complexity of the proposed algorithm is independent of the size of the state space. While this is true, it is mostly a result of restrictive assumptions on the problem rather than the algorithmic contributions of the paper and the statements can be misleading. The problem is assumed to have a fixed start state, a finite horizon, and transition dynamics that are close to a deterministic simulator. These three conditions limit the number of states that often appear in trajectories, which is independent of the size of the total state space. In fact replacing the fixed start state with a set of possible start states (in the appendix) brings the cardinality of this starting set into the sample complexity. More generally if the starting distribution is over all the states then the size of the state space will appear in the sample complexity.

3. One of the functions approximators is assumed to be large enough to satisfy a realizability assumption (Assumption 3). It is said that the assumption can be satisfied with a deep neural network but the size of this function class appears as a logarithmic factor in the sample complexity. Is this factor finite for a deep neural network and does it depend on the size of the state space or the space of observations?

Typos:
- line 93: q(s|x) -> q(x|s)
- line 134: "at eta-perturbation" -> "an eta-perturbation"
- line 179: instantiating

--------------------------------------
Update after discussions:

Thanks for the clarifications. I increased the score to 7 as the comments 1 and 3 are addressed. I'll elaborate on the second comment:

The problem setting in this paper has two important properties. First, the environment is nearly deterministic in terms of eta-perturbation so that a nearly accurate deterministic simulator exists in the first place. Second, this simulator is available to the agent. The algorithm achieves sample complexity independent of |S| in this problem setting (which is great). Now the reader may want to know if both of these properties are essential to this sample complexity or if the first one is enough. That is, if the environment is nearly deterministic in terms of eta-perturbation, can an algorithm without access to a simulator or without learning a simulator achieve this sample complexity in a non-transfer setting? I understand that achieving this low sample complexity in a general stochastic problem is impossible, but is it also impossible in this _nearly_ deterministic environment? Also I think running some model-free baselines on the mini-grid experiment and seeing if their sample complexity grows with |S| is a good idea.

Another minor comment: Other reviews initially raised doubt about the significance of this result. I suggest referencing or bringing a lower bound from previous work that grows with |S| without the assumptions in this paper. This should better motivate this result and highlight that requiring some assumptions is inevitable for achieving this sample complexity. I can think of Theorem 2 in [1] right now but the authors have a better grasp on this topic and may find a more relevant bound.

[1] Dann, Christoph, and Emma Brunskill. "Sample complexity of episodic fixed-horizon reinforcement learning." Advances in Neural Information Processing Systems 28 (2015).

---

> ### Author Response · Authors · 2022-08-02
> **Author Response: Major clarification on non-triviality of avoiding dependence on the size of state space**
>
> Thank you for the feedback. We comment on the three weaknesses described in the review in the same order below:
>
> 1. **Previous Work and No Simulator:** We are not aware of other algorithms in our setting that achieve a sample complexity result that is independent of the size of state space. Block MDPs have been extensively studied over the last several years, however, all Block MDP algorithms have a dependence on state space, and in the worst case, it cannot be avoided. These algorithms do not assume access to a simulator and cannot be easily extended to yield a solution that avoids dependence on state space.  In contrast, our work employs a novel decoding approach that learns a non-Markovian decoder by relying on learning an inverse dynamics model and using it together with the simulator.
>
>      If the deterministic simulator is not given but one exists, then it can be learned from the source environment. This can be accomplished by running an algorithm like HOMER (Misra et al., 2020) that can extract latent dynamics for a Block MDP. Note that we only care about sample cost in the target environment as they are more expensive in practice, therefore, the sample cost of running Homer, in this case, will be 0 as we don't use the target environment.
>
>
> 2. **Major Clarification:** Even though our simulator is deterministic, due to perturbations in the target environment, a policy can visit every state in the target environment with non-zero probability. Thus, all $|S|$ states can be reached by the same policy where $|S|$ can be as large as $|A|^H$. Therefore, our result avoiding $|S|$ dependence is non-trivial.
>
>
> 3. **Log of the Size of Model Class**: For neural networks, the set of all models is basically the set of all possible weight vectors, so if the weight vector is represented by a vector of size 1000, where each entry is encoded in double precision (i.e., with 64 bits), then the $\log_2$ of the number of models is 64,000. Alternatively, the generalization bound (Theorem 4) that is stated in terms of the log of the size of model class, can also be easily generalized to Rademacher complexity. Note that generalization bounds like these are routinely used in reinforcement learning analysis with function approximation (e.g., see Jiang et al., 2017, Du et al., 2019, Misra et al., 2020, Agarwal et al., 2020). Our experiments on the visual grid world show that neural networks indeed can solve this task reliably well in practice.

---

> ### Author Response · Authors · 2022-08-08
> **Thank you**
>
> We thank the reviewer for their time and effort. As the discussion period ends tomorrow, we would be happy to explain anything further or address more questions. If we were able to address your questions and concerns, then we would appreciate if you can update your review. Thanks again for putting the time into reviewing our paper.

---

### Official Review · Reviewer_Rkjz · 2022-07-12

**Rating:** 6
**Confidence:** 4
**Soundness:** 3 good
**Presentation:** 3 good
**Contribution:** 2 fair

**Summary:**

This paper studies a transfer learning setting, where the agent wishes to learn in a block MDP target environment with the help of an approximated latent simulator. This paper proposed an algorithm that achieves sample efficient learning in such a setting given that
(1) the latent simulator is deterministic while still achieving a small approximation error. In other words, the target environment is nearly deterministic.
(2) the learner must know the approximation error of the simulator or at least a tight upper bound.
(3) the learner has access to a realizable function class of the block MDP non-latent dynamics.

**Questions:**

1. What is the technical barrier that requires the deterministic latent transition assumption? Overall I feel the proposed approach can sometimes be very suboptimal. If the real environment has high stochasticity, forcing a deterministic approximation and then looking for a robust policy to the approximation error doesn't seem to be a good idea, since the resulting suboptimality gap will be very large.

2. The learner requires the knowledge of the approximation error of the simulator, which further weakens the problem setting and makes it even less practical.

3. Why does the algorithm attempt to learn a non-Markovian decoder when the ground-truth decoder is in fact Markovian?

4. While the paper is only concerned with block MDPs, it makes a rather strong model-based realizability assumption (Assum. 3) which has been shown to be unnecessary for block MDPs, e.g. [1]. This assumption seems to be needed mainly to use the inverse dynamics model techniques. It's not obvious to me whether such an assumption, along with (1) deterministic simulator and (2) known approximation error are necessary or not for the particular setting this paper considers. For example, one could potentially apply the representation learning subroutine in [1] and then call the robust latent policy on top of the learned representation for exploration, which is a much simpler approach both conceptually and computationally.

[1] Zhang, Xuezhou, Yuda Song, Masatoshi Uehara, Mengdi Wang, Wen Sun, and Alekh Agarwal. "Efficient reinforcement learning in block mdps: A model-free representation learning approach."

Overall, I am most concerned with the strong assumptions required for the proposed approach to work. Without further justification, e.g. lower bounds, it's hard to appreciate the theoretical contribution of this paper.


**Limitations:**

the assumptions are not adequately discussed and justified.

**Strengths And Weaknesses:**

The setting studied in this paper appears to be new, at least in the theoretical literature, and has some practical motivation. However, the assumptions required by the proposed algorithm to work is too strong comparing to recent literature on sample efficient learning in block MDPs.

---

> ### Author Response · Authors · 2022-08-02
> **Author Response: Major clarification on Zhang et al.,**
>
> Thank you for your detailed feedback. We first make an important clarification regarding Zhang et al., and then discuss the other remarks.
>
> 1. **Major Clarification:** We make three important clarifications regarding Zhang et al.
>
> -    The cited paper of Zhang et al., makes a realizability assumption. Specifically, they assume that their decoder class contains $\phi^\star$. We are happy to explain this point in greater detail if desired. This is analogous to our assumption where we assume that the correct model for the inverse dynamics lies in our function class. In general, realizability assumptions are very common in observation-based reinforcement learning (e.g., Du et al., 2019, Misra et al., 2020, Agarwal et al., 2020, Mhammedi et al., 2020). Further, a few agnostic results that exist yield pessimistic results with exponential sample complexity (e.g., “Agnostic Reinforcement Learning with Low-Rank MDPs and Rich Observations”, Dann et al.).
>
> - Sample complexity of Zhang et al., depends on the rank for low-rank MDPs which implies it depends on the size of state space for Block MDPs. Therefore, applying their routine will give us dependence on state space which we avoid with our novel solution. In real-world problems, the state space can be combinatorially large and, therefore, our solution offers an exponential speed-up when our assumptions hold.
>
> - We also note that Zhang et al., learn a representation by solving a min-max optimization problem which is more difficult to solve in practice, than learning an inverse dynamics model which reduces to simple multi-class classification. Further, inverse dynamics learning is very commonly used in empirical research. E.g., see "Curiosity-driven Exploration by Self-supervised Prediction", Pathak et al, ICML 2017 and the recent "Video PreTraining (VPT): Learning to Act by Watching Unlabeled Online Videos", Baker et al., 2022.
>
>
> 2. **Regarding "deterministic assumption":** The place where we use the determinism of the simulator is in our decoding strategy, where after inferring the _shadow actions_, we play the inferred action sequence in the simulator to find the final state (Algorithm 2, Line 12). If the simulator is stochastic, then there is no unique final state upon executing an action sequence.
>
>      Our main motivation behind "near-deterministic setting" stems from its application in robotics, where such assumptions are relatively common and which is also a crucial application area for sim-to-real or transfer RL settings. Our assumptions also capture sticky-actions which are common in deep RL settings.
>
>     Certainly, near-deterministic assumptions are inapplicable to certain problems, and this restriction is not made for general block MDP settings (Du et al., 2019, Misra et al., 2020), however, these algorithms have sample complexity that depends on the size of state space, a dependence that we crucially remove. Studying whether one can avoid dependence on state space while transferring from stochastic simulator, is an interesting future work direction. However, we believe that our work contains novel and interesting results with interesting set of applications and, therefore, stands on its own.
>
> 3. **Upper bound on $\eta$:** TASSID does assume access to an upper bound on the perturbation. This assumption can be relaxed by a standard halving trick. We can first assume the maximum perturbation, i.e., $\eta = ½$, and run TASSID. This will be overly conservative when there is less perturbation. We can estimate the value of the learned policy in the target environment. We then run TASSID with $\eta=¼$, and repeat the process. We stop halving when the value of the final policy stops improving. We expect a procedure like this to converge in $O(\log 1/\eta_{true})$ rounds in practice where $\eta_{true}$ is the smallest valid perturbation noise.
>
> 4. **Non-Markovian Decoder**: While the current observation has sufficient information to decode the current state, we were unable to find any procedure that can decode the current state using just the current observation without introducing sample complexity dependence on either the state space or exponential in the horizon. Crucially, directly applying existing Block MDP algorithms doesn't give the desired guarantee and we get cascading errors. Our novel decoding mechanism extracts the current state by retracing the path that the agent has followed in the simulator. This requires access to all observations in the past. We view this as an interesting property of our solution.
>
>     We would also note that non-Markovian decoders have been used in the past for *rich-observation* control problems (LQR) using a different learning algorithm (see Line 28, Algorithm 5 in “Learning the Linear Quadratic Regulator from Nonlinear Observations”, Mhammedi et al., NeurIPS 2020).

---

> > ### Comment · Reviewer_Rkjz · 2022-08-07
> > **Score Updated.**
> >
> > Thank you for the detailed explanation. Now I see that the main focus here is to get dimension-independent sample complexity and prior approaches such as [1] are not able to obtain that. I have adjusted my rating accordingly.
> >
> > Nevertheless, I would still encourage the authors to think harder about the necessity of the assumptions being made in the paper. For example, can you construct a hard example showing that non-deterministic MDPs cannot be learned with a dimension-free sample complexity? Such results would complement the upper bounds nicely and greatly strengthen the paper.

---

### Author Response · Authors · 2022-08-09
**Updated the paper based on reviews**

We thank all reviewers for their time and useful feedback. Based on the feedback, we have updated the paper to

- Add a table of notations in the Appendix

- Fix a typo and add some clarification in the proof (Reviewer 3Ver)

We will be happy to incorporate any additional suggestions to improve the manuscript.

---

### Meta-Review · Area_Chair_33Uj · 2022-08-26

**Recommendation:** Accept
**Confidence:** Less certain

**Metareview:**

This paper studies how to improve learning efficiency in settings where the agent has access to an abstract, simplified model of the world.

All the reviewers voted to accept and each pointed out useful clarifications and improvements to the paper. The main contributions of this work are the novel problem setting, algorithms proposed and theoretical results. There are some proof of concept experiments included as well.

The paper is not without flaw though:
- the intro dismisses two related works because those algorithms were "very different". This is either a communication problem or a bad excuse
- the proof of concept experiments don't really match the motivation of the paper well. The toy environments are not representative of the targeted complexity (this is not to say they must be messy, high-dim real sensor data.)

Most importantly the paper has basic correctness issues with the experiments provided:
- reporting results from five runs (seeds) in toy domains and reporting standard deviation (which is likely not well estimated from 5 runs) instead use a more conservative measure of confidence such as student-t CI or bootstrap intervals
- no clear discussion of how the baselines where tuned (they all use the same step-size in each experiment. With Adam the step-size is less sensitive, but its very unlikely that the values of alpha used (and other hypers) were best. We need a clear description of the empirical methodology used here; researcher descent is a very biased process and not good enough
--> This is particularly problematic because the abstract claims the new method outperforms the baselines

These empirical issues are common practice in the field but poor none the less. It is very likely all could be addressed and the main messages of the paper continue to hold. The empirical writeup of this work significantly weakens an otherwise strong paper.

**Award:**

No

---

### Decision · Program_Chairs · 2022-09-14

Accept